# RETHINKING OUTPUT ALIGNMENT FOR 1-BIT POST-TRAINING QUANTIZATION OF LARGE LANGUAGE MODELS

## ABSTRACT

Large Language Models (LLMs) deliver strong performance across a wide range of NLP tasks, but their massive sizes hinder deployment on resource-constrained devices. To reduce their computational and memory burden, various compression techniques have been proposed, including quantization, pruning, and knowledge distillation. Among these, post-training quantization (PTQ) is widely adopted for its efficiency, as it requires no retraining and only a small dataset for calibration, enabling low-cost deployment. Recent advances for post-training quantization have demonstrated that even sub-4-bit methods can maintain most of the original model performance. However, 1-bit quantization that converts floating-point weights to $\pm 1$, remains particularly challenging, as existing 1-bit PTQ methods often suffer from significant performance degradation compared to the full-precision models. Specifically, most of existing 1-bit PTQ approaches focus on weight alignment, aligning the full-precision model weights with those of the quantized models, rather than directly aligning their outputs. Although the output-matching approach objective is more intuitive and aligns with the quantization goal, naively applying it in 1-bit LLMs often leads to notable performance degradation. In this paper, we investigate why and under what conditions output-matching fails, in the context of 1-bit LLM quantization. Based on our findings, we propose a novel data-aware PTQ approach for 1-bit LLMs that explicitly accounts for activation error accumulation while keeping optimization efficient. Empirical experiments demonstrate that our solution consistently outperforms existing 1-bit PTQ methods with minimal overhead.

## 1 INTRODUCTION

Large language models (LLMs) (Wei et al., 2022; Radford et al., 2019b; Zhang et al., 2022; Brown et al., 2020b) have become a focal point of both academic research and industrial development, thanks to their strong capabilities across a wide range of natural language processing tasks (Hendrycks et al., 2020; Bisk et al., 2020b), including question answering (Devlin et al., 2019), machine translation (Fan et al., 2020; Lepikhin et al., 2020), summarization (Zhang et al., 2019; Lewis et al., 2019) and language generation (Radford et al., 2019a; Brown et al., 2020a). Despite these advances, the massive scale of modern LLMs, which often involving billions of parameters, poses substantial challenges for efficient inference and deployment. To address this, the community has explored various compression approaches, such as neural architecture search (Zoph & Le, 2016), knowledge distillation (Hinton et al., 2015), network quantization (Choi et al., 2018; Frantar et al., 2023), and pruning (Han et al., 2015). However, many of these approaches depend on large-scale training data and costly retraining, which limits their practicality. In contrast, post-training quantization (PTQ) (Liu et al., 2025; Sun et al., 2025) requires only a small calibration set and modest computational resources, making it a practical choice for compressing LLMs. Despite the impressive performance of sub-4-bit PTQ methods, the most extreme case, 1-bit quantization, remains challenging, which maps floating-point parameters to binary states, and can greatly lower memory consumption.

Existing 1-bit quantization approaches can be broadly grouped into two categories: (1) *weight-matching methods*, which minimize $\|W - \widehat{W}\|$ (referred as the **Weight Error**, *i.e.*, the distance

between full-precision weights $W$ and binarized weights $\widehat{W}$) (Xu et al., 2018; Shang et al., 2023), and (2) *output-matching methods*, which minimize $\|\widehat{X}W - \widehat{X}\widehat{W}\|$ (referred to as **Activation-conditioned Error**, since it compares outputs given the same quantized model's layer inputs $\widehat{X}$ (Li et al., 2024).

In the context of LLM quantization, the primary objective is to align the outputs of the quantized model with those of the full-precision model. Weight-matching methods, which minimize $\|W - \widehat{W}\|$, are simple and stable but do not directly optimize the output-alignment objective. Despite recent advances, most 1-bit PTQ techniques remain weight-centric (Huang et al., 2024; Li et al., 2024; Dong et al., 2024; Shang et al., 2023). ARB-X (Li et al., 2024) is an exception, which incorporates **Activation-conditioned Error**, *i.e.*, minimizing $\|\widehat{X}W - \widehat{X}\widehat{W}\|$. However, ARB-X has two primary limitations. Firstly, it naively applies output alignment in a layer-wise manner, which does not guarantee improvement at the block-level or at the final output due to inter-layer dependencies. Secondly, by conditioning on $\widehat{X}$ rather than the true full-precision input $X$, the objective $\|\widehat{X}W - \widehat{X}\widehat{W}\|$ is only an approximation; as quantization errors accumulate across layers, the approximated target outputs $W\widehat{X}$ diverge from the true full-precision target $WX$, reducing the effectiveness of layer-wise output alignment in PTQ.

Motivated by the above analysis, in this paper, we propose a selective layer-wise output matching method to ensure block-level loss reduction. Our objective explicitly accounts for accumulated quantization errors by directly matching the output of the quantized model with the true target output **Output Error**, *i.e.*, $\|WX - \widehat{W}\widehat{X}\|$. Furthermore, we observe that the effectiveness of output alignment is architecture-dependent: indiscriminate application can significantly degrade attention mechanisms, particularly in architectures such as LLaMA. To mitigate this issue, we introduce a novel masking mechanism, termed Attention Matrix Preservation (AMP), which preserves attention behavior and prevents performance degradation. These design choices collectively yield a simple yet effective data-aware 1-bit quantization strategy for LLMs.

The main contributions of this paper can be summarized as follows:

- We systematically examine the influence of calibration data on 1-bit post-training quantization for LLMs, revealing the insight that while output matching aligns with the quantization objective, its effectiveness can vary depending on model architecture and layer characteristics.
- Our study identifies three key challenges in naive layer-wise output alignment: (i) it does not necessarily reduce block-level loss, (ii) quantization errors accumulate across layers, diminishing alignment effectiveness, and (iii) indiscriminate output matching can disrupt token interactions, degrading attention mechanisms, particularly in LLMs.
- To address these challenges, we propose a selective layer-wise output alignment strategy that modifies the quantization objective to explicitly account for accumulated errors. Moreover, we also introduce an attention-aware masking mechanism AMP to preserve attention behavior.
- Extensive experiments demonstrate that our method consistently improves performance over existing 1-bit PTQ techniques for LLMs.

## 2 RELATED WORKS

**Quantization in LLMs.** Post-Training Quantization (PTQ) has emerged as the most practical strategy for compressing large language models (LLMs), as it applies quantization directly to pre-trained models with minimal calibration data, avoiding the prohibitive cost of Quantization-Aware Training (QAT). A range of PTQ methods have been developed to mitigate quantization error including GPTQ (Frantar et al., 2023) that leverages second-order Hessian information for layer-wise error compensation; AWQ (Lin et al., 2023) and SmoothQuant (Xiao et al., 2023) that incorporate activation statistics to identify and preserve critical weights; and ZeroQuant (Yao et al., 2022) that introduces fine-grained schemes for improved flexibility. More recent efforts such as QuIP (Tseng et al., 2024) and QuaRot (Ashkboos et al., 2024) extend PTQ with rotation or vector quantization to better distribute outliers, though often at the expense of higher computational overhead. Collectively, these efforts have helped LLMs maintain strong performance under moderate precision settings (e.g., 4–8 bits), yet the models still suffer from substantial degradation when pushed to extreme regimes such as 1-bit quantization.

**1-Bit Quantization for Language Languages Models.** Binarization, where weights are restricted to $\pm 1$, represents the most aggressive form of quantization. It was first explored in computer vision with specialized binary architectures such as XNOR-Net (Rastegari et al., 2016) and Bi-Real Net (Liu et al., 2018), which showed that binary parameters could still capture meaningful representations. Follow-up studies (Guo et al., 2017; Xu et al., 2018) improved 1-bit quantization through enhanced coding schemes and optimized search strategies, enabling more accurate approximations of full-precision weights. Inspired by these advances, recent work has extended binarization to LLMs. Training-based approaches, such as BitNet (Wang et al., 2023), demonstrated that end-to-end training with binary weights is feasible. In contrast, post-training quantization (PTQ) approaches aim to binarize pretrained models with minimal retraining. BiLLM (Huang et al., 2024) selectively quantizes salient weights with low-bit precision while binarizing the rest, guided by Hessian-based importance and residual-aware masks. STB-LLM (Dong et al., 2024) combines pruning and quantization with fine-grained grouping, achieving sub-1-bit average precision while maintaining accuracy, albeit with added kernel and storage costs. Other methods leverage codebook representations to capture repeating binary patterns, improving compression without requiring sparsity. Most recently, research has shifted toward data-aware and fine-grained quantizers tailored for 1-bit PTQ. ARB (Li et al., 2024) introduces grouping and refinement strategies to reduce quantization error, and its data-aware extension ARB-X further optimize the output alignment.

## 3 PRELIMINARY ANALYSIS

In the following, we provide a preliminary analysis of how data and output alignment affect 1-bit LLM quantization. Although **Activation-Conditioned Error** is more aligned with the quantization objective, most existing 1-bit PTQ for LLMs approaches instead try to minimize **Weight Error** during the quantization process. We aim to understand why output alignment is less widely adopted, and why naive output alignment does not necessarily improve model performance.

### 3.1 EFFECT OF LAYER OUTPUT MATCHING ON BLOCK-LEVEL PERFORMANCE

Quantization objectives are typically formulated at the layer-wise, block-wise, or network-wise level. Prior work such as BRECQ (Li et al., 2021) has shown that block-wise quantization is particularly effective, since layers within the same block are highly interdependent. This suggests that minimizing the error at the block level is more critical than focusing solely on individual layers.

To assess the impact of layer-wise output matching on block-level loss, we conduct a preliminary analysis using ARB and ARB-X Li et al. (2024). ARB performs layer-wise weight alignment by minimizing the **Weight Error** $||W - \widehat{W}||$, whereas ARB-X extends this to layer-wise output alignment, *i.e.*, the **Activation-conditioned Error**. The evaluation is performed on the LLaMA-2-7B model using the C4 calibration set. For each transformer block, we measure the block-level output loss when applying ARB or ARB-X to an individual layer while keeping all other layers in the block at full precision, as illustrated in Fig. 1. Notably, some layers show higher block-level loss under ARB-X compared to ARB, despite ARB-X reducing the corresponding layer-level loss. This result demonstrates that naive layer-wise output alignment does not necessarily improve block-level performance relative to weight alignment, revealing a fundamental limitation of ARB-X and its output matching.

### 3.2 IMPACT OF ACCUMULATED QUANTIZATION ERROR ON OUTPUT ALIGNMENT

We next study how accumulated error affects the quantized model over time, which can influence the effectiveness of the output alignment objective. To do so, we leverage ARB-X (Li et al., 2024) as the baseline for output alignment analysis, and evaluate it on the Llama-2-7B model using C4 calibration sets. We evaluate two types of errors: (1) **Activation-conditioned Error**, *i.e.*, $||\widehat{X}W - \widehat{X}\widehat{W}||$, which is the objective of ARB-X (Li et al., 2024), and (2) **Output Error**, *i.e.*, $||XW - \widehat{X}\widehat{W}||$, the discrepancy between the quantized and full-precision layer outputs. In parallel, we present cosine similarity measures, by replacing the MSE loss with the cosine similarity, denoted as (1) **Activation-conditioned Similarity** and (2) **Output Similarity**. Both metrics are measured in a block-wise manner across all 32 blocks of the architecture during the quantization process.

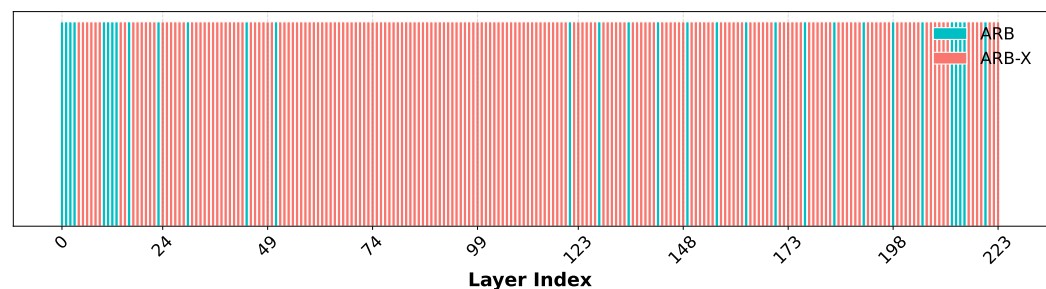

Figure 1: Comparison of block-level loss under ARB (weight alignment) versus ARB-X (layer-wise output alignment) for each layer of LLaMA-2-7B. Bar color denotes which method incurs lower block loss: blue for ARB, red for ARB-X. Several layers demonstrate that minimizing layer-level loss via ARB-X does not necessarily achieve lower block-level loss than ARB, indicating the limitations of naive layer-wise output alignment.

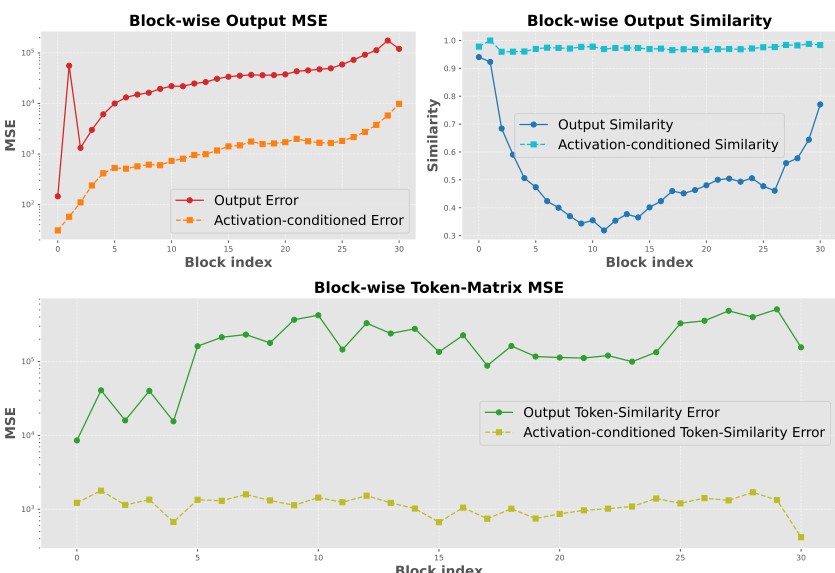

Figure 2: Accumulated quantization error in LLaMA-2-7B under ARB-X. The top plot reports cosine similarity and MSE between block outputs of the quantized and full-precision models. While ARB-X maintains low cosine similarity loss between pre- and post-quantization outputs, the MSE relative to the full-precision baseline grows with depth, indicating error accumulation. The bottom plot reports the MSE loss for token similarity matrices, computed as pairwise cosine similarities between token representations within each sample. These matrices also drift from the full-precision baseline after quantization, suggesting that ARB-X's output alignment may degrade the learned attention mask in subsequent layers.

As shown in the upper-right panel of Figure 2, ARB-X maximizes the cosine similarity between the layer outputs before and after the quantization of that layer, $\widehat{X}W$ and $\widehat{X}\widehat{W}$. However, the mean squared error (MSE) remains substantial, and the cosine similarity with the actual full-precision output $XW$ decreases throughout the quantization process. This illustrates the limitation of naive output alignment: as quantization errors accumulate across layers, the optimization objective progressively deviates from the true target, thereby diminishing its effectiveness.

## 3.3 EFFECT OF OUTPUT MATCHING ON ATTENTION MECHANISM

The growing discrepancy in layer outputs suggests that token-to-token interactions, which underlie attention patterns, may be affected during quantization. To investigate this, we extend the evaluation protocol described in Section 3.2 and analyze the Llama-2-7B model using ARB-X. For

each sample $\widehat{X}_i$, we compute token similarity matrices as $\widehat{X}_i \widehat{W} \widehat{W}^\top \widehat{X}_i^\top$, after row-normalizing $\widehat{X}_i \widehat{W}$, so that entry $(j, k)$ represents the cosine similarity between tokens $j$ and $k$ in the quantized layer output. For methods minimizing **Activation-conditioned Error**, the target similarity matrix is $\widehat{X}_i W W^\top \widehat{X}_i^\top$, while for methods minimizing **Output Error**, it is $X_i W W^\top X_i^\top$. These matrices serve as a proxy for the attention mask learned by the model. We quantify deviations using (1) **Activation-conditioned token-similarity error**, defined as $\sum_i \left\| \widehat{X}_i \widehat{W} \widehat{W}^\top \widehat{X}_i^\top - \widehat{X}_i W W^\top \widehat{X}_i^\top \right\|$, and (2) **Output token-similarity error**, defined as $\sum_i \left\| \widehat{X}_i \widehat{W} \widehat{W}^\top \widehat{X}_i^\top - X_i W W^\top X_i^\top \right\|$.

This evaluation captures how well the quantized model preserves self-attention interactions. As shown in the lower part of Figure 2, the token similarity matrices gradually diverge from the full-precision baseline as depth increases. These results suggest that naive output alignment frameworks such as ARB-X may inadvertently distort attention masks in deeper layers, thereby weakening the token-level relational structure. This arises because, when the output matching loss remains large, the optimization tends to prioritize reducing errors along high-magnitude channels. While this reduces Euclidean distance, it often comes at the expense of preserving the directional alignment of representations, ultimately degrading similarity across tokens.

By focusing on token-level interactions, this insight complements the previous analysis: it highlights that output alignment should be designed with awareness of attention patterns to better preserve the learned token relationships in deeper layers.

## 4 METHOD

In this section, we present our data-aware quantization strategy for 1-bit post-training quantization (PTQ) of large language models (LLMs). Our design is motivated by three key observations from the preliminary analysis: (i) layer-wise output matching does not necessarily lead to block-level loss reduction, (ii) activation mismatches can accumulate across layers, and (iii) naive output alignment may disrupt token interactions, thereby degrading the attention mask. To address these issues, our strategy (a) applies output matching selectively at the block level, (b) modifies the quantization objective to account for accumulated errors, and (c) introduces attention-aware adjustments to preserve attention behavior.

Consider a neural network with $L$ layers, trained with a loss function $\ell$ on a calibration dataset of size $n$. Let $W \in \mathbb{R}^{d_{\mathrm{in}} \times d_{\mathrm{out}}}$ denote the full-precision weight matrix and $\widehat{W}$ its quantized version. Given the full-precision layer input $X \in \mathbb{R}^{n \times d_{\mathrm{in}}}$, the full-precision layer output is $Z = XW$. In the quantized model, the input $\widehat{X}$ denotes the activations produced after quantizing all previous $l - 1$ layers, and the corresponding layer output of the quantized model is $\widehat{Z} = \widehat{X} \widehat{W}$.

Most PTQ methods for 1-bit LLMs minimize the weight alignment loss for the layer $l$ as follows:

$$\mathcal{L}(X, l) = \| W - \widehat{W} \|_F^2, \tag{1}$$

where $\|.\|_F$ denotes the Frobenius norm. ARB-X (Li et al., 2024), a recent PTQ method, proposes to minimize the layer-wise output reconstruction error for the layer indexed by $l$ as follows:

$$\mathcal{L}(X, l) = \left\| \widehat{X} W - \widehat{X} \widehat{W} \right\|_F^2 = \mathrm{Tr} \left[ (W - \widehat{W})^\top S (W - \widehat{W}) \right], \tag{2}$$

where $\widehat{S} = \widehat{X}^\top \widehat{X}$ is the Gram matrix of the quantized activations. However, this objective does not take into account the accumulation error of the quantization process from prior layers. Therefore, we modify the optimization objective, by adopting the full-precision input $X$ for the target output as follows:

$$\mathcal{L}(X, l) = \left\| XW - \widehat{X} \widehat{W} \right\|_F^2 = \mathrm{Tr} \left[ (XW - \widehat{X} \widehat{W})(XW - \widehat{X} \widehat{W})^\top \right]. \tag{3}$$

We following a similar strategy in ARB-RC (Li et al., 2024) to parameterize the quantized model weight $\widehat{W} = \mathrm{diag}(\alpha_r) B \mathrm{diag}(\alpha_c)$, where $B \in \{-1, 1\}^{d_{\mathrm{in}} \times d_{\mathrm{out}}}$, $\alpha_r \in \mathbb{R}^{d_{\mathrm{in}}}$ and $\alpha_c \in \mathbb{R}^{d_{\mathrm{out}}}$ and

diag(.) denotes the diagonal matrix. The optimization objective can then be written as follows:

$$\mathcal{L}(X, L) = \|f_Q(X) - f_{FP}(X)\|_F^2$$

$$= \mathrm{Tr}\left[(XW - \widehat{X}\widehat{W})(XW - \widehat{X}\widehat{W})^\top\right] \tag{4}$$

$$= \mathrm{Tr}\left[(XW - \widehat{X}\,\mathrm{diag}(\alpha_r)B\mathrm{diag}(\alpha_c))(XW - \widehat{X}\,\mathrm{diag}(\alpha_r)B\mathrm{diag}(\alpha_c))^\top\right].$$

We have three parameters to optimize, including $\alpha_r$, $\alpha_c$ and $B$. Regarding the parameter $\alpha_c$, we can obtain its optimal closed-form by setting the gradient of $\alpha_c$ to 0. The optimal solution for $\alpha_c$ can be derived as follows:

$$\alpha_c^* = \frac{\mathrm{Diag}(B^\top \mathrm{diag}(\alpha_r)SW)}{\mathrm{Diag}(B^\top \mathrm{diag}(\alpha_r)\widehat{S}\mathrm{diag}(\alpha_r)B)} \tag{5}$$

with $S = \widehat{X}^\top X$, and $\mathrm{Diag}(.)$ denotes the diagonal vector of the input.

For the binary matrix $B$, as it has binary constraint, we cannot get its optimal solution by setting the gradient of the objective loss $B$ to 0. However, inspired by (Shen et al., 2015), we can derive the optimal closed-form solution for each row $i$ in $B$ while keeping other rows of $B$ fixed. Let $N = \mathrm{diag}(\alpha_r)S\mathrm{diag}(\alpha_r)$, $K = \mathrm{diag}(\alpha_c \odot \alpha_c)$ and $P = \mathrm{diag}(\alpha_c)W^\top S\mathrm{diag}(\alpha_r)$. Each row of $B$ then has the optimal closed-form solution as follows:

$$B_{i,:}^* = \mathrm{sign}(N_F BK - 2P)_{i,:}, \tag{6}$$

where $N_F = N - \mathrm{diag}(\mathrm{diagonal}(N))$ is the matrix $N$ but its diagonal is set to 0.

Regarding the parameter $\alpha_r$, we approximate its closed-form solution by solving the following:

$$(\widehat{S} \odot C)\alpha_r = \mathrm{Diag}(SW\mathrm{diag}(\alpha_c)B^\top), \tag{7}$$

where $C = B\,\mathrm{diag}(\alpha_c \odot \alpha_r)\,B^\top$. This yields the closed-form expression.

$$\alpha_r^* = (\widehat{S} \odot C)^{-1}\mathrm{Diag}(SW\mathrm{diag}(\alpha_c)B^\top), \tag{8}$$

where $(\widehat{S} \odot C)^{-1}$ denotes the Moore–Penrose pseudoinverse. In practice, directly computing the pseudoinverse can be numerically unstable. Instead, we employ the `torch.linalg.lstsq` function to obtain a stable least-squares solution. Full derivations for all variables are provided in Appendix B.

## 4.1 ATTENTION MATRIX PRESERVATION

As demonstrated in Section 3.3, LLM architectures such as Llama witness significant degradation in the attention masks when using output alignment. In order to mitigate this problem during the quantization process, we propose a novel Attention Matrix Preservation (AMP) mechanism, that avoids the degradation of the attention masks. Specifically, the token-similarity matrix of the model's output at a layer $l$ of input $\widehat{X}$ is defined as $\widehat{X}\widehat{W}\widehat{W}^\top\widehat{X}^\top$ after normalizing $\widehat{X}\widehat{W}$. Similarly, the token-similarity matrix of the full precision output is denoted as $XWW^\top X^\top$. Since the attention mask is closely correlated with the similarity matrix across tokens, the objective to minimize the attention degradation problem is defined as:

$$\max \mathcal{L}_{AMP} = \left\|(\widehat{X}\widehat{W}\widehat{W}^\top\widehat{X}^\top) \odot (XWW^\top X^\top)\right\|$$

$$= \mathrm{Tr}\left[\widehat{X}\widehat{W}\widehat{W}^\top\widehat{X}^\top XWW^\top X^\top\right]$$

$$= \mathrm{Tr}\left[\widehat{W}^\top \underbrace{\widehat{X}^\top XWW^\top X^\top \widehat{X}}\, \widehat{W}\right]$$

$$= \mathrm{Tr}\left[\widehat{W}^\top \qquad M \qquad \widehat{W}\right] \tag{9}$$

For each quantization parameter $\alpha_c$, $\alpha_r$ and $B$, we assign them an AMP mask, defined as the sign of the gradient of $\mathcal{L}_{AMP}$ w.r.t. these parameters:

$$M^c = AMP(\alpha_c) = \mathrm{sign}(\mathrm{Diag}(B^\top \mathrm{diag}(\alpha_r)M\widehat{W}))$$

$$M^r = AMP(\alpha_r) = \mathrm{sign}(\mathrm{Diag}(M\widehat{W}\mathrm{diag}(\alpha_c)B^\top)) \tag{10}$$

$$M^B = AMP(B) = \mathrm{sign}(\mathrm{diag}(\alpha_r)M\widehat{W}\mathrm{diag}(\alpha_c))$$

In order to avoid the degradation of the token similarity matrix, once we obtain the AMP mask for each variable $\alpha_c$, $\alpha_r$ and $B$, we update them with:

$$\alpha_r = \alpha_r * (1 - M^r) + \alpha_r^* * M^r$$
$$\alpha_c = \alpha_c * (1 - M^c) + \alpha_c^* * M^c \tag{11}$$
$$B_{i,:} = B_{i,:} * (1 - M_{i,:}^B) + B_{i,:}^* * M_{i,:}^B$$

### 4.2 FINAL OPTIMIZATION

Using the closed-form solution above, we jointly optimize all three variables until convergence. However, as shown in Section 3.1, naively applying output alignment to all layers within a block does not necessarily minimize the block-level loss. To address this, we adopt a selective layer-wise output approach, by restricting the output alignment to only the last fully connected layer of each block, since it has the most direct impact on the block loss, while employing weight alignment methods such as ARB-RC (Li et al., 2024) for quantizing the remaining layers of each block. Our complete algorithm is provided in Algorithm 1 (see Appendix E).

## 5 EXPERIMENTS

In this section, we conduct extensive experiments to validate the effectiveness and superiority of our proposed method compared to current SOTA 1-bit LLM quantization frameworks.

### 5.1 SETUP

**Models and datasets.** Our experiments are conducted on the OPT (Zhang et al., 2022), covering parameter scales from 1.3B up to 30B, and LLaMA model families, including LLaMA-2 (Touvron et al., 2023) and the recently released LLaMA-3 (Dubey et al., 2024). We do not include LLaMA-1 in our evaluation since the original pretrained checkpoints are not officially available through Hugging Face or other standard model hubs. For evaluation, we adopt widely used benchmarks in prior 1-bit LLM quantization works. Perplexity is reported on WikiText2 (Merity et al., 2016), PTB (Marcus et al., 1993), and C4 (Raffel et al., 2020), which are standard for measuring language modeling quality. To further assess downstream capability, we also measure zero-shot performance on seven QA datasets: ARC-Easy and ARC-Challenge (Clark et al., 2018), PIQA (Bisk et al., 2020a), BoolQ (Clark et al., 2019), HellaSwag (Zellers et al., 2019), WinoGrande (Sakaguchi et al., 2021) and OBQA (Mihaylov et al., 2018). Additionally, we also provide the zero-shot performance of our method on Llama models, adding LAMBADA Paperno et al. (2016) for long-context reasoning evaluation. Please refer to the Appendix for the zero-shot results on Llama models.

**Baseline methods.** We compare our method against several state-of-the-art 1-bit PTQ methods, including BiLLM (Huang et al., 2024), ARB-LLM(Li et al., 2024) and PB-LLM (Shang et al., 2023), ensuring that all implementations adhere to the details provided in their respective papers. BiLLM (Huang et al., 2024), ARB-LLM(Li et al., 2024) and PB-LLM (Shang et al., 2023) all utilize the PTQ approach for model calibration through OBQ based method of GPTQ. For ARB-LLM, we evaluate two of its best performing variants, ARB-X and ARB-RC. The ARB-RC results in Tables 1 and 2 were obtained by running the original ARB-RC implementation.

### 5.2 EXPERIMENTAL RESULTS

**Results on Language Generation Tasks.** We evaluate our method in terms of perplexity for both OPT and LLaMA models. Table 1 presents perplexity results for OPT models across the C4 and WikiText-2 datasets, including OPT-1.3B, OPT-2.7B, OPT-6.7B, OPT-13B, and OPT-30B. Table 1 also reports the average accuracy of our method on seven zero-shot QA datasets for OPT models. For LLaMA models, Table 2 reports results for LLaMA-2-7B, LLaMA-2-13B, and LLaMA-3-8B. Our method consistently outperforms previous state-of-the-art quantization approaches across all benchmarks. Notably, for more challenging settings such as OPT-1.3B and OPT-2.7B, we achieve up to 4.85 and 3.42 reductions in perplexity, highlighting the robustness and effectiveness of our approach. For the performance of the method on Llama models, we achieve from 0.22-2.22 reduction across benchmarks, with the exception of Llama-2-7B model evaluated on PTB dataset.

Table 1: Comparison of our method with different 1-bit quantization methods for OPT models. Perplexity (↓) is reported for C4, WikiText2, and PTB, while Accuracy (↑) is reported for Zero-shot QA datasets. Alignment Type denotes if the method use Weight Alignment (WA) or Output Alignment (OA)

| Dataset | Metric | Method | Alignment Type | | Block Size | Weight Bits | 1.3B | 2.7B | 6.7B | 13B | 30B |
|---|---|---|---|---|---|---|---|---|---|---|---|
| | | | WA | OA | | | | | | | |
| | | Full Precision | - | - | - | 16 | 16.07 | 14.34 | 12.71 | 12.06 | 11.45 |
| | | PB-LLM | ✓ | | 128 | 1.7 | 168.12 | 222.15 | 104.78 | 57.84 | 27.67 |
| C4 | PPL (↓) | BiLLM | ✓ | | 128 | 1.11 | 64.14 | 44.77 | 42.13 | 19.83 | 16.17 |
| | | ARB-RC | ✓ | | 128 | 1.11 | 27.70 | 21.46 | 16.97 | 15.07 | 13.34 |
| | | ARB-X | | ✓ | 128 | 1.11 | 47.60 | 34.97 | 22.54 | 17.71 | 14.71 |
| | | **Ours** | | ✓ | 128 | 1.11 | **24.69** | **19.90** | **16.22** | **14.71** | **13.15** |
| | | Full Precision | - | - | - | 16 | 14.62 | 12.47 | 10.86 | 10.13 | 9.56 |
| | | PB-LLM | ✓ | | 128 | 1.7 | 239.81 | 278.27 | 144.25 | 74.59 | 28.30 |
| WikiText2 | PPL (↓) | BiLLM | ✓ | | 128 | 1.11 | 69.05 | 48.61 | 47.65 | 18.75 | 13.86 |
| | | ARB-RC | ✓ | | 128 | 1.11 | 26.40 | 19.84 | 14.92 | 13.10 | 11.19 |
| | | ARB-X | | ✓ | 128 | 1.11 | 45.40 | 34.37 | 20.07 | 15.47 | 12.36 |
| | | **Ours** | | ✓ | 128 | 1.11 | **24.30** | **18.25** | **14.56** | **12.84** | **10.94** |
| | | Full Precision | - | - | - | 16 | 20.29 | 17.97 | 15.77 | 14.52 | 14.04 |
| | | PB-LLM | ✓ | | 128 | 1.7 | 324.62 | 183.97 | 169.49 | 101.00 | 41.87 |
| PTB | PPL (↓) | BiLLM | ✓ | | 128 | 1.11 | 115.94 | 88.52 | 69.41 | 27.16 | 21.41 |
| | | ARB-RC | ✓ | | 128 | 1.11 | 43.03 | 31.77 | 22.31 | 19.09 | 16.88 |
| | | ARB-X | | ✓ | 128 | 1.11 | 71.96 | 54.28 | 31.23 | 23.46 | 19.28 |
| | | **Ours** | | ✓ | 128 | 1.11 | **38.18** | **28.35** | **21.45** | **18.85** | **16.75** |
| | | PB-LLM | ✓ | | 128 | 1.7 | 36.60 | 37.06 | 35.95 | 37.40 | 43.70 |
| AveQA | Acc. (↑) | BiLLM | ✓ | | 128 | 1.11 | 38.89 | 40.44 | 38.27 | 47.00 | 49.61 |
| | | ARB-RC | ✓ | | 128 | 1.11 | 45.22 | 48.25 | 52.58 | 55.01 | 57.11 |
| | | ARB-X | | ✓ | 128 | 1.11 | 40.52 | 42.21 | 46.57 | 49.19 | 51.77 |
| | | **Ours** | | ✓ | 128 | 1.11 | **45.76** | **49.03** | **53.33** | **55.06** | **57.70** |

Table 2: Perplexity (↓) of LLaMA-2 and LLaMA-3 models under different quantization methods for C4, WikiText2, and PTB datasets. Alignment Type denotes if the method use Weight Alignment (WA) or Output Alignment (OA)

| Dataset | Method | Alignment Type | | Block Size | Weight Bits | LLaMA-2 | | LLaMA-3 |
|---|---|---|---|---|---|---|---|---|
| | | WA | OA | | | 7/8B | 13B | 8B |
| | Full Precision | - | - | - | 16 | 7.26 | 6.73 | 9.45 |
| | PB-LLM | ✓ | | 128 | 1.7 | 80.69 | 184.67 | 104.15 |
| C4 | BiLLM | ✓ | | 128 | 1.06 | 39.38 | 25.87 | 61.04 |
| | ARB-RC | ✓ | | 128 | 1.06 | 20.4 | 14.77 | 36.04 |
| | ARB-X | | ✓ | 128 | 1.06 | 28.02 | 19.82 | 41.86 |
| | **Ours** | | ✓ | 128 | 1.06 | **19.25** | **13.8** | **35.14** |
| | Full Precision | - | - | - | 16 | 5.47 | 4.88 | 6.14 |
| | PB-LLM | ✓ | | 128 | 1.7 | 66.41 | 236.40 | 73.08 |
| WikiText2 | BiLLM | ✓ | | 128 | 1.06 | 32.31 | 21.35 | 55.80 |
| | ARB-RC | ✓ | | 128 | 1.06 | 16.25 | 12.47 | 27.42 |
| | ARB-X | | ✓ | 128 | 1.06 | 21.61 | 14.86 | 31.98 |
| | **Ours** | | ✓ | 128 | 1.06 | **15.42** | **11.5** | **27.20** |
| | Full Precision | - | - | - | 16 | 37.91 | 50.93 | 11.18 |
| | PB-LLM | ✓ | | 128 | 1.7 | **657.24** | 816.31 | 106.25 |
| PTB | BiLLM | ✓ | | 128 | 1.06 | 5243.01 | 309.12 | 87.25 |
| | ARB-RC | ✓ | | 128 | 1.06 | 763.19 | 197.70 | 47.88 |
| | ARB-X | | ✓ | 128 | 1.06 | 681.24 | 182.10 | 53.86 |
| | **Ours** | | ✓ | 128 | 1.06 | 3166 | **196.64** | **45.66** |

However, the large perplexity indicates that the metric cannot provide a meaningful evaluation. For the evaluation on QA datasets, our method consistently outperforms all other methods, up to 0.78% improvement.

Table 3: Layer-wise ablation (AMP) for LLaMA-2-7B and OPT-6.7B.

| Model / Mtd. | | PPL (↓) | |
|---|---|---|---|
| | | C4 | WikiText2 |
| **LLaMA-2-7B** | No AMP | 29.12 | 26.24 |
| | AMP | 19.25 | 15.42 |
| **OPT-6.7B** | No AMP | 16.35 | 14.74 |
| | AMP | 16.22 | 14.56 |

Table 4: Ablation on activation/output error objectives for LLaMA-2-7B and OPT-6.7B.

| Model / Obj | | PPL (↓) | |
|---|---|---|---|
| | | C4 | WikiText2 |
| **LLaMA-2-7B** | Act. Error | 19.97 | 15.66 |
| | Out. Error | **19.25** | **15.42** |
| **OPT-6.7B** | Act. Error | 16.91 | 14.83 |
| | Out. Error | **16.22** | **14.74** |

### 5.3 ABLATION STUDY

To analyze the effectiveness of our proposed method, we perform ablation experiments on OPT and Llama models. Please refer to our Appendix for more ablation studies and results.

**Impact of activation accumulation error.** To investigate the impact of accumulated error on model performance and assess the effectiveness of our method, we conduct an ablation study where we optimize our method using the **Activation-conditioned Error** (the same objective as ARB-X Li et al. (2024)) instead of the **Output Error**. The results are presented in Table 4. As shown, explicitly accounting for accumulated error in our optimization objective yields a 0.7 improvement in perplexity on the C4 dataset.

**Impact of Attention Matrix Preservation.** To evaluate the impact of our proposed Attention Matrix Preservation (AMP) on model performance, we conduct an ablation study comparing settings with and without AMP (Table 3). Figure 3 in the Appendix visualizes the token similarity matrices of LLaMA-2-7B under our method using the C4 calibration set. Overall, model performance degrades for both OPT and LLaMA models without AMP. Notably, LLaMA suffers severe degradation, with perplexity increasing by over 10 points, indicating that its token similarity deteriorates more than in OPT. We hypothesize that this sensitivity arises because LLaMA uses RMSNorm instead of LayerNorm: RMSNorm normalizes each token to unit norm before applying a learned scale, making the model more dependent on the direction of representations and therefore more vulnerable to quantization-induced deviations. AMP plays a key role in mitigating this degradation by preserving the token similarity structure, which helps maintain the integrity of attention patterns during quantization.

**Overhead Analysis.** Please refer to Appendix D

## 6 CONCLUSION

In this work, we investigated the role of calibration data in 1-bit post-training quantization of large language models. Our analysis revealed important insights: layer-wise output matching does not necessarily reduce block-level error; activation mismatches can accumulate across layers; and naive output alignment may degrade attention masking, all of which can negatively impact the effectiveness of output matching for 1-bit post-training quantization. These findings provide a deeper understanding of the limitations of existing PTQ objectives and constitute a contribution on their own. Building on these insights, we introduced a quantization strategy that selectively applies output alignment at the block level, incorporates attention-aware masking, and reformulates the quantization objective to account for accumulated error. Extensive experiments demonstrate that our method consistently outperforms prior 1-bit PTQ approaches for LLMs.

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

## A  APPENDIX

**THE USE OF LARGE LANGUAGE MODELS**    We used a large language model (ChatGPT) to help with editing this paper. It was only used for simple tasks such as fixing typos, rephrasing sentences for clarity, and improving word choice. All ideas, experiments, and analyses were done by the authors, and the use of LLMs does not affect the reproducibility of our work.

## B  DERIVATION OF CLOSED-FORMS

**Derivation for $\alpha_c$**    The gradient of the objective w.r.t $\alpha_c$ is:

$$
\begin{aligned}
\frac{\partial \mathcal{L}}{\partial \mathrm{diag}(\alpha_c)} &= \frac{\partial \mathcal{L}}{\partial(XW - \widehat{X}\widehat{W})}\frac{\partial(XW - \widehat{X}\widehat{W})}{\partial \alpha_c} \\
&= -2 * B^\top \mathrm{diag}(\alpha_r)\widehat{X}^\top(XW - \widehat{X}\widehat{W}) \\
&= -2 * B^\top \mathrm{diag}(\alpha_r)\widehat{X}^\top(XW - \widehat{X}\mathrm{diag}(\alpha_r)B\mathrm{diag}(\alpha_c))
\end{aligned}
\tag{12}
$$

Setting the diagonal of this gradient to 0, we have:

$$
\begin{aligned}
\mathrm{diagonal}(B^\top \mathrm{diag}(\alpha_r)\widehat{S}\mathrm{diag}(\alpha_r)B) \odot \alpha_c &= \mathrm{diagonal}(B^\top \mathrm{diag}(\alpha_r)SW) \\
\implies \alpha_c &= \frac{\mathrm{diagonal}(B^T \mathrm{diag}(\alpha_r)SW)}{\mathrm{diagonal}(B^T \mathrm{diag}(\alpha_r)\widehat{S}\mathrm{diag}(\alpha_r)B)}
\end{aligned}
\tag{13}
$$

**Derivation for $\alpha_r$**

$$
\begin{aligned}
\frac{\partial \mathcal{L}}{\partial \mathrm{diag}(\alpha_r)} &= \frac{\partial \mathcal{L}}{\partial(XW - \widehat{X}\widehat{W})}\frac{\partial(XW - \widehat{X}\widehat{W})}{\partial \alpha_r} \\
&= -2 * \widehat{X}^\top(XW - \widehat{X}\widehat{W})\mathrm{diag}(\alpha_c)B^\top \\
&= -2 * \widehat{X}^\top(XW - \widehat{X}\mathrm{diag}(\alpha_r)B\mathrm{diag}(\alpha_c))\mathrm{diag}(\alpha_c)B^\top
\end{aligned}
\tag{14}
$$

Setting the diagonal of this gradient to 0, we have:

$$
\begin{aligned}
\mathrm{diagonal}(\widehat{S}\mathrm{diag}(\alpha_r)\underbrace{B\mathrm{diag}(\alpha_c \odot \alpha_c)B^\top}) &= \mathrm{diagonal}(SW\mathrm{diag}(\alpha_c)B^\top) \\
\implies \mathrm{diagonal}(\widehat{S}\mathrm{diag}(\alpha_r)\qquad \mathcal{B}\qquad) &= \mathrm{diagonal}(SW\mathrm{diag}(\alpha_c)B^\top) \\
\implies (\widehat{S} \odot \mathcal{B})\alpha_r &= \mathrm{diagonal}(SW\mathrm{diag}(\alpha_c)B^\top)
\end{aligned}
\tag{15}
$$

**Derivation for $B$**    Explaining the objective loss $\mathcal{L}(.)$ out we have:

$$
\begin{aligned}
\mathcal{L}(X, L) &= \mathrm{Tr}\left[(XW - \widehat{X}\widehat{W})(XW - \widehat{X}\widehat{W})^T\right] \\
&= \mathrm{Tr}\left[\widehat{X}\widehat{W}\widehat{W}^\top\widehat{X}^\top - 2 * XW\widehat{W}^\top\widehat{X}^\top + \mathrm{const}\right] \\
&\propto \mathrm{Tr}\left[\widehat{X}^\top\widehat{X}\widehat{W}\widehat{W}^\top\right] - 2 * \mathrm{Tr}\left[\widehat{X}^\top XW\widehat{W}^\top\right] \\
&= \mathrm{Tr}\left[\widehat{S}\widehat{W}\widehat{W}^\top\right] - 2 * \mathrm{Tr}\left[SW\widehat{W}^\top\right] \\
&= \mathrm{Tr}\left[\underbrace{\mathrm{diag}(\alpha_r)S\mathrm{diag}(\alpha_r)}B\underbrace{\mathrm{diag}(\alpha_c \odot \alpha_c)}B^\top\right] - 2 * \mathrm{Tr}\left[\underbrace{\mathrm{diag}(\alpha_c)SW\mathrm{diag}(\alpha_r)}B^\top\right] \\
&= \mathrm{Tr}\left[\qquad N\qquad B\quad K\quad B^\top\right]\qquad - 2 * \mathrm{Tr}\left[\qquad P\qquad B^\top\right]
\end{aligned}
\tag{16}
$$

We denote $\widetilde{B}_{-i}$ and $\widetilde{N}_{-i}$ respectively as the matrix $B$ and $N$ exclude the row $i$. Similarly, $\widetilde{N}_{i,-j}$ denotes the row i of matrix $N$ excluded the $j$ element. Given a row $i$ of $B$, keeping the other row of

$B$ as constant, the loss can be expanded as a function of $B_{i,:}$ as:

$$\mathcal{L}_{MSE}(X, L) \propto \text{Tr}\left[NBKB^\top\right] - 2 * \text{Tr}\left[PB^\top\right]$$

$$= \sum_i (N_{i,:}BKB_{i,:}^\top - 2P_{i,:}B_{i,:}^\top)$$

$$= \sum_i (N_{i,i}B_{i,:}KB_{i,:}^\top + \widetilde{N}_{i,-i}\widetilde{B}_{-i}KB_{i,:}^\top - 2P_{i,:}B_{i,:}^\top) \quad (17)$$

$$\propto \sum_i (\widetilde{N}_{i,-i}\widetilde{B}_{-i}K - 2P_{i,:})B_{i,:}^\top$$

Since $B_{i,:}$ is a binary vector, it has a closed-form solution:

$$B_{i,:} = \text{sign}(\widetilde{N}_{i,-i}\widetilde{B}_{-i}K - 2P_{i,:}) \quad (18)$$

The more compacted form of the whole $B$ using this closed-form for each row can be computed as:

$$B = \text{sign}(N_F BK - 2P) \quad (19)$$

with $N_F = N - \text{diag}(\text{diagonal}(N))$ is the matrix $N$ setting its diagonal to 0.

## C  VISUALIZATION OF THE IMPACT OF AMP MASK TO ATTENTION MECHANISM DEGRADATION

We provide visualization of the impact our AMP mask in mitigating the attention degradation problem of output alignment.

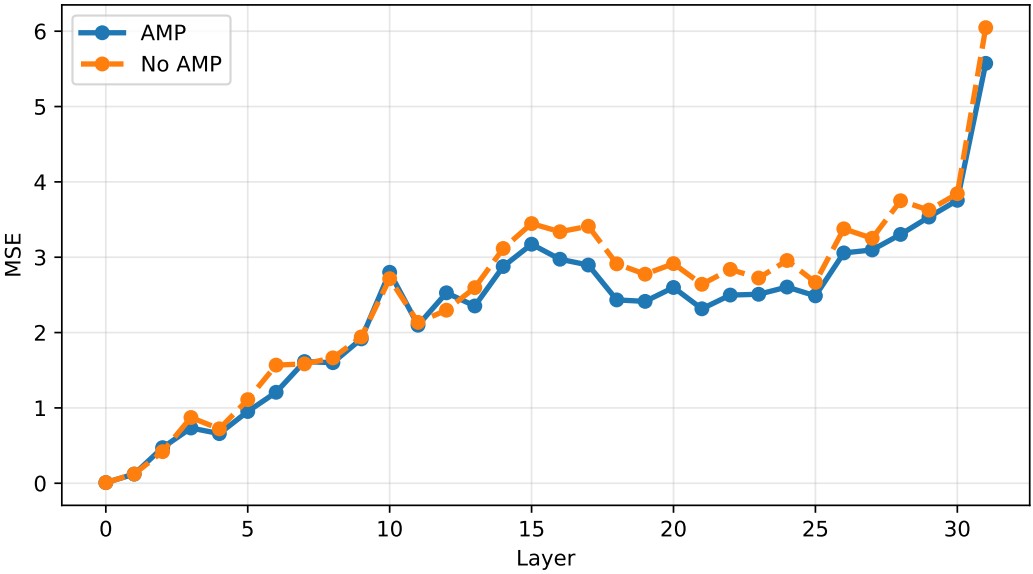

Figure 3: Block-wise MSE reconstruction error between quantized and full-precision attention score in LLaMA-2-7B. Two curves represent the errors of quantized models with and without the AMP mask, computed over C4 calibration data.

## D  ADDITIONAL RESULTS

**Additional Details of Experiments.**  All experiments are implemented in PyTorch and executed on a single NVIDIA GeForce RTX A100 GPU. Consistent with prior studies such as GPTQ (Frantar et al., 2023) and BiLLM (Huang et al., 2024), we use the C4 dataset with a sequence length of 2048 as calibration data to enable fair comparison. The quantization block size is fixed at 128 following ARB (Li et al., 2024)

Table 5: Ablation study of applying our method to different layers (Q, K, V, Out, Final) of each block for LLaMA-2-7B and OPT-6.7B, evaluated on C4 and WikiText2.

| Model / Layer | | PPL ($\downarrow$) | |
|---|---|---|---|
| | | C4 | WikiText2 |
| Llama-2-7B | Query | 20.08 | 15.85 |
| | Key | 20.80 | 16.75 |
| | Value | 21.44 | 17.82 |
| | Attn Out | 21.02 | 18.11 |
| | **Final FC** | **19.25** | **15.42** |
| OPT-6.7B | Query | 17.15 | 15.15 |
| | Key | 17.13 | 15.06 |
| | Value | 17.12 | 14.81 |
| | Attn Out | 17.05 | 15.55 |
| | **Final FC** | **16.22** | **14.56** |

**Ablation study for output alignment of different layers.**   To investigate the impact of output alignment at different layers on model performance, we conduct an ablation study applying our method to individual layers within each block of OPT and Llama models. The results are presented in Table 5. As observed, output alignment is most effective and consistent when applied to the final layer of each block, for both Llama and OPT models.

**Inference and Storage Overhead Analysis.** Our method introduces **no additional inference or storage overhead**, as it does not add any new quantization parameters and leaves both the model architecture and forward-pass computations unchanged. Consequently, the memory footprint and runtime during inference are identical to ARB-RC. As reported in ARB (Li et al., 2024), ARB-RC achieves similar inference time to BiLLM (Huang et al., 2024), $4.3$–$4.6\times$ faster than PB-LLM (Shang et al., 2023) and $4.4$–$5.1\times$ faster than the full-precision model, hence these performance gains also apply to our method.

**Quantization Overhead.**   We provide in detail the quantization time of our method, compared to ARB-X and ARB-RCLi et al. (2024), across architecture. While our method incurs slightly higher overhead than ARB-RC due to the additional closed-form computations and AMP mask, it remains more efficient than ARB-X. Importantly, post-training quantization for LLMs is already highly efficient; for example, quantizing Llama-2-13B requires only about two hours on a single A100 GPU. Since quantization is a one-time process, this modest overhead is practically negligible and does not affect inference speed or deployment efficiency.

Table 6: Quantization time comparison between 1-bit LLM methods and ours across different models.

| Model | ARB-X | ARB-RC | Ours |
|---|---|---|---|
| **OPT-6.7B** | 87m | 65m | 90m |
| **Llama-2-7B** | 91m | 54m | 73m |
| **Llama-2-13B** | 147m | 100m | 116m |

**Additional zero-shot QA results for LLama models.**   We provide additional results of our method using 8 different zero-shot QA datasets, over Llama architectures

Table 7: Evaluation the accuracy ($\uparrow$) of LLaMA models using ARB-RC and our method over zero-shot QA benchmarks.

| Model Size | Method | Bits | Boolq | Lambada | Piqa | OPQA | Winogrande | Arc-E | Arc-C | Hellaswag | Avg |
|---|---|---|---|---|---|---|---|---|---|---|---|
| **LLaMA-2-7B** | ARB-RC | 1.06 | 67.74 | 51.87 | 65.51 | 29.8 | 59.98 | 46.8 | 28.24 | 48.1 | 49.75 |
| | **Ours** | 1.06 | 66.45 | 52.53 | 68.12 | 29.8 | 56.99 | 30.3 | 51.56 | 49.24 | **50.62** |
| **LLaMA-2-13B** | ARB-RC | 1.06 | 74.86 | 66.23 | 71.16 | 33.0 | 62.04 | 57.37 | 33.36 | 50.46 | 56.06 |
| | **Ours** | 1.06 | 72.02 | 68.99 | 72.58 | 36.0 | 63.14 | 59.85 | 33.7 | 53.92 | **57.53** |
| **LLaMA-3-8B** | ARB-RC | 1.06 | 67.58 | 48.85 | 62.73 | 29.2 | 57.38 | 42.97 | 24.57 | 43.63 | 47.11 |
| | **Ours** | 1.06 | 66.54 | 49.35 | 63.33 | 30.4 | 56.2 | 45.66 | 26.02 | 44.21 | **47.71** |

**Ablation study for the hyper-parameter** $k$ We provide additional ablation study for the hyper-parameter $k$ in Table 8. In practice, we adopt $k = 5$ for stable performance across architecture.

Table 8: Ablation study for the hyper-parameter $k$.

| Model | k | PPL $\downarrow$ | |
|---|---|---|---|
| | | C4 | WikiText2 |
| OPT-6.7B | 1 | 16.24 | 14.67 |
| | 5 | 16.22 | 14.56 |
| | 10 | 16.21 | 14.40 |
| LLaMA2-7B | 1 | 20.16 | 16.26 |
| | 5 | 19.25 | 15.42 |
| | 10 | 20.00 | 16.42 |

# E PROPOSED LEARNING ALGORITHM

---
**Algorithm 1** Our-RC
---
1: **procedure** OUR-RC($W, S, \widehat{S}, T, k$)
2:    $\triangleright W \in \mathbb{R}^{d_{in} \times d_{out}}$: *full-precision weight*      $\triangleleft$
3:    $\triangleright S \in \mathbb{R}^{d_{in} \times d_{in}}$: *Gram matrix* $X^\top \widehat{X}$      $\triangleleft$
4:    $\triangleright \hat{S} \in \mathbb{R}^{d_{in} \times d_{in}}$: *Gram matrix* $\widehat{X}^\top \widehat{X}$      $\triangleleft$
5:    $\triangleright T$: *iteration rounds*      $\triangleleft$
6:    Initialize $\widehat{W}, \alpha_r, \alpha_c, B$
7:    Initialize $M \leftarrow SWW^T S^T$
8:    **for** iter = 1:$T$ **do**
9:      $\alpha_r^* \leftarrow$ REFINE-$\alpha_r(S, \widehat{S}, W, B, \alpha_c, M)$
10:      $M^r \leftarrow$ AMP-$\alpha_r(M, B, \alpha_r, \alpha_c)$
11:      $\alpha_r \leftarrow \alpha_r * (1 - M^r) + \alpha_r^* * M^r$
12:      **if** $(\text{iter} + 1) \bmod k == 0$ **then**
13:        $\alpha_c^* \leftarrow$ refine-$\alpha_c(S, \widehat{S}, W, B, \alpha_r)$
14:        $M^c \leftarrow$ AMP-$\alpha_c(M, B, \alpha_r, \alpha_c)$
15:        $\alpha_c \leftarrow \alpha_c * (1 - M^c) + \alpha_c^* * M^c$
16:      $B^* \leftarrow$ REFINE-B($S, \widehat{S}, W, B, \alpha_c$)
17:      $M^B, i \leftarrow$ AMP-B($M, B, \alpha_r, \alpha_c$)
18:      $B_{i,:} \leftarrow B_{i,:} * (1 - M_{i,:}^B) + B_{i,:}^* * M_{i,:}^B$
19:      $B^* \leftarrow$ REFINE-B($S, \widehat{S}, W, B, \alpha_c$)
20:      $M^B, i \leftarrow$ AMP-B($M, B, \alpha_r, \alpha_c$)
21:      $B_{i,:} \leftarrow B_{i,:} * (1 - M_{i,:}^B) + B_{i,:}^* * M_{i,:}^B$
22:    **return** $\widehat{W}$

**Algorithm 2** Auxiliary functions

1: **procedure** REFINE-$\alpha_r(S, \widehat{S}, W, B, \alpha_c, M)$
2:     $\mathcal{B} = B \operatorname{diag}(\alpha_c \odot \alpha_r) B^\top$
3:     matrix $:= \widehat{S} \odot \mathcal{B}$
4:     target $:= SW \operatorname{diag}(\alpha_c) B^\top$
5:     $\alpha_r \leftarrow$ SOLVE(matrix, target)
6:     **return** $\alpha_r$

7: **procedure** REFINE-$\alpha_c(S, \widehat{S}, W, B, \alpha_r)$
8:     num $:= \operatorname{Diag}(B^\top \operatorname{diag}(\alpha_r) SW)$
9:     den $:= \operatorname{Diag}(B^\top \operatorname{diag}(\alpha_r) \widehat{S} \operatorname{diag}(\alpha_r) B)$
10:     $\alpha_c :=$ num/den
11:     **return** $\alpha_c$

12: **procedure** REFINE-B$(S, \widehat{S}, W, \alpha_r, \alpha_c)$
13:     $N = \operatorname{diag}(\alpha_r) S \operatorname{diag}(\alpha_r)$
14:     $K = \operatorname{diag}(\alpha_c \odot \alpha_c)$
15:     $P = \operatorname{diag}(\alpha_c) W^\top S \operatorname{diag}(\alpha_r)$
16:     $\mathbf{B} = N_F B K - 2P$

17:     $B^* = \operatorname{sign}(B)$
18:     $i \leftarrow \operatorname{argmax}_j \sum_k (B \odot \mathbf{B})_{j,k}$
19:     **return** $B^*, i$

20: **procedure** AMP_B$(M, B, \alpha_r, \alpha_c)$
21:     $\widehat{W} = \operatorname{diag}(\alpha_r) B \operatorname{diag}(\alpha_c)$
22:     $M^B = \operatorname{sign}(\operatorname{diag}(\alpha_r) M \widehat{W} \operatorname{diag}(\alpha_c))$
23:     **return** $M^B$

24: **procedure** AMP-$\alpha_r(M, B, \alpha_r, \alpha_c)$
25:     $\widehat{W} = \operatorname{diag}(\alpha_r) B \operatorname{diag}(\alpha_c)$
26:     $M^r = \operatorname{sign}(\operatorname{Diag}(M \widehat{W} \operatorname{diag}(\alpha_c) B^\top))$
27:     **return** $M^r$

28: **procedure** AMP-$\alpha_c(M, B, \alpha_r, \alpha_c)$
29:     $\widehat{W} = \operatorname{diag}(\alpha_r) B \operatorname{diag}(\alpha_c)$
30:     $M^c = \operatorname{sign}(\operatorname{Diag}(B^\top \operatorname{diag}(\alpha_r) M \widehat{W}))$
31:     **return** $M^c$

