# OpenReview forum: "Rethinking Output Alignment for 1-bit Post-Training Quantization of Large Language Models"
_ICLR.cc/2026/Conference — ICLR 2026 Conference Withdrawn Submission_

### Official Review · Reviewer_kLf3 · 2025-10-17

**Soundness:** 2
**Presentation:** 2
**Contribution:** 2
**Rating:** 2
**Confidence:** 5

**Summary:**

This work combines ARB-X and ARB-RC, partially addressing ARB-RC's inability to aware calibration data through approximate solutions. Furthermore, the authors revise the quantization objective to make the quantized output more closely match that of the full-precision version. In addition, they incorporate an AMP mask into the updated formula to prevent performance decline in the attention mechanism. Finally, the authors apply their approach to the last fully connected layer of each block, while keeping ARB-RC in all other layers. Experiments demonstrate that this approach attains a marginal improvement over ARB-RC.

**Strengths:**

1. The authors provide a thorough analysis of the errors introduced by quantization, revealing that smaller layer-wise errors do not necessarily translate into smaller block-wise errors, which more accurately reflect the model's ultimate performance.
2. The article provides a detailed derivation of the formulas, with rigorous notation.

**Weaknesses:**

1. The final performance improvement compared to the previous state-of-the-art method was only 0.78%, raising concerns about the effectiveness of the method.
2. The proposed method was only applied to the last fully connected layer in each transformer block; all other layers still used ARB-RC. In the ablation experiments in Table 5, the authors' method performed even worse than ARB-RC when applied to layers other than the final FC layer, demonstrating the limitations of their proposed method.

**Questions:**

1. The analysis of the impact of quantization error on the attention mechanism in Section 3.3 is confusing. The true attention mechanism's token similarity is calculated by mapping token representations using two different matrices, W_q and W_k . The authors simplify this analysis by reducing the calculation of token similarity to the result of a single matrix mapping. Is this a reasonable approximation? Does it reflect the impact of quantization on the true attention mechanism?
2. The ablation experiments on AMP in Table 3 show that the update method without AMP even leads to larger errors than ARB-RC, particularly on LLaMA-2-7B. Do the authors have a reasonable explanation for this?
3. Does the hyperparameter k in Table 8 correspond to the number of iterations? More explanation is needed to clarify this.
4. Regarding the time cost of quantization, do the authors use 5 iterations for their method, while the comparison method uses 15? Perhaps a comparison using the same number of iterations would be more reasonable. Since their method is much more complex than ARB-RC, the speedup achieved by quantization (Table 6) is confusing.

---

> ### Author Response · Authors · 2025-11-25
>
> # Reviewer kLf3
>
> We greatly appreciate the time and effort the Reviewer dedicated to considering our paper. Here are our responses to all concerns raised by the Reviewer.
>
>
>
> > **W1**: The final performance improvement compared to the previous state-of-the-art method was only 0.78%, raising concerns about the effectiveness of the method.
>
> **Answer**:
> Our method does not introduce new quantization parameters, change the quantization scheme, or add meaningful overhead compared to ARB-X or ARB-RC. Despite this minimal modification, it consistently improves perplexity by up to 4.85 points and yields 0.7–1.47% average accuracy gains across multiple zero-shot QA benchmarks, which is substantial.
> These gains are achieved solely by improving the closed-form update formulation in the final layer of each block, a lightweight modification that can be integrated into any 1-bit PTQ framework. Beyond raw performance , the paper also contributes new insights explaining why naive output alignment is rarely adopted, offering a foundation for future work (e.g., exploring which additional layers within each block can also safely adopt output alignment)
>
> ---
>
> > **W2**: The proposed method was only applied to the last fully connected layer in each transformer block; all other layers still used ARB-RC. In the ablation experiments in Table 5, the authors' method performed even worse than ARB-RC when applied to layers other than the final FC layer, demonstrating the limitations of their proposed method.
>
> **Answer**:
> This question assumes that our output-alignment objective is intended to be applied across all layers. However, the first insight of our paper in Section 3.1 is precisely the opposite:
> **naively applying output alignment to every layer is neither effective nor theoretically justified, as it can disrupt the block-level reconstruction loss.**  Within each transformer block, only the final FC layer of each block is guaranteed to have a direct and stable influence on the block output under quantization.   Other layers may also benefit from output alignment, but not uniformly.
> For this reason, our method intentionally applies Eq. 3 + AMP only to the final FC layer, where output alignment is actually meaningful. A promising future direction is to explore whether a small non-uniform subset of intermediate layers beside last layer of each block may also benefit from output alignment, rather than applying the objective uniformly for only last layer of each block.
>
> ---
>
> > **Q1**: The analysis of the impact of quantization error on the attention mechanism in Section 3.3 is confusing. The true attention mechanism's token similarity is calculated by mapping token representations using two different matrices, W_q and W_k. The authors simplify this analysis by reducing the calculation of token similarity to the result of a single matrix mapping. Is this a reasonable approximation? Does it reflect the impact of quantization on the true attention mechanism?
>
> **Answer**:
> In the revision, we include the revised visualization for Figure 3 comparing the true attention scores of full-precision models with those produced by the quantized model. These results empirically show that output alignment (without AMP) noticeably degrades the actual attention pattern, confirming that our simplified analysis captures the core failure reason. The approximation is therefore sufficient to reflect the qualitative impact of quantization on attention.
>
> ---
>
> > **Q2**: The ablation experiments on AMP in Table 3 show that the update method without AMP even leads to larger errors than ARB-RC, particularly on LlaMA-2-7B. Do the authors have a reasonable explanation for this?
>
> **Answer**:
> Please see General Response Q4
>
> ---
>
> > **Q3**: Does the hyperparameter k in Table 8 correspond to the number of iterations? More explanation is needed to clarify this.
>
> **Answer**:
> In our optimization, we update three quantized parameters: $\alpha_r$, $\alpha_c$, and $B$.
> The hyperparameter $k$ does *not* denote the total number of iterations; instead, it controls the *update frequency* of $\alpha_c$.
> Concretely, $\alpha_r$ and $B$ are updated at every iteration, while $\alpha_c$ is updated once every $k$ iterations. Empirically, we observe that $\alpha_c$ changes much more slowly than $\alpha_r$ and $B$, so updating it less frequently reduces computation without affecting convergence.

---

> ### Author Response · Authors · 2025-11-25
>
> ---
>
> > **Q4**: Regarding the time cost of quantization, do the authors use 5 iterations for their method, while the comparison method uses 15? Perhaps a comparison using the same number of iterations would be more reasonable. Since their method is much more complex than ARB-RC, the speedup achieved by quantization (Table 6) is confusing.
>
> **Answer**:
> Our method uses the *same* total number of optimization steps as ARB-RC. Specifically, we run 15 iterations in total: $\alpha_r$ and $B$ are updated at every iteration, and $\alpha_c$ is updated once every 5 iterations within these 15 steps (as explained in Q3). Thus, the comparison is made under an identical iteration budget.
>
> In terms of complexity, our approach is not substantially more expensive than ARB-RC: we keep the same quantization scheme and only modify the closed-form update rules for the quantized variables. Usually, data-aware optimisation is often slower than data-free method. In our method, the proposed data-aware output-alignment objective is applied *only* to the final layer of each transformer block, while all other layers still use the cheaper ARB-RC update. As a result, our method retains efficiency close to ARB-RC, and remains significantly faster than ARB-X, which applies data-dependent optimization more broadly.

---

> > ### Comment · Reviewer_kLf3 · 2025-11-26
> >
> > Thank the authors for their response.
> >
> > Regarding W1, I would like to emphasize that ARB/ARB-X does not introduce new quantization parameters nor alter the quantization scheme compared to the prior work BiLLM. Nevertheless, it achieves a significantly more substantial improvement, which reduces the perplexity by 28 compared to the previous SOTA method on the Llama-7B model. In contrast, the performance gain brought by the proposed method in this paper is considerably more modest than that of preceding works.
> >
> > Regarding Q1, there is a discrepancy between Figure 3 and its caption. The figure displays MSE, while the caption refers to "attention mass." Based on the current visualization, it is difficult to establish a clear connection between AMP and attention mechanisms.
> >
> > Regarding W2 and Q2, the authors acknowledge that the proposed output alignment method is only applicable to fc2 layers. Furthermore, using Equation 3 alone still leads to performance degradation. Eq 3 must be combined with AMP to yield any improvement, which remains marginal. This indicates that the approach proposed in this paper requires careful tuning to achieve marginal benefits. Additionally, although the authors provide some potential explanations, they lack enough evidence and a unified framework to systematically account for these phenomena.

---

> ### Author Response · Authors · 2025-11-26
>
> > **Q1**: Regarding W1, I would like to emphasize that ARB/ARB-X does not introduce new quantization parameters nor alter the quantization scheme compared to the prior work BiLLM. Nevertheless, it achieves a significantly more substantial improvement, which reduces the perplexity by 28 compared to the previous SOTA method on the Llama-7B model. In contrast, the performance gain brought by the proposed method in this paper is considerably more modest than that of preceding works.
>
>
> We strongly disagree with the premise that ARB method does not introduce additional parameters or modify the quantization structure relative to BiLLM.
> - **Firstly**, ARB-X introduces an additional *refined mean* variable, which BiLLM does not include. This increases the optimization freedom and directly expands the optimisation space.
> - **Secondly**, ARB-RC further introduces *per-column scaling parameters*, which again increase the number of learnable quantization parameters beyond those in BiLLM.
> - **Thirdly**, the ARB family employs a more fine-grained *group bitmap* strategy, partitioning weights into more (and smaller) groups than BiLLM. This results in more optimization parameters and greater flexibility.
>
> Thus, ARB-X/ARB-RC **do** introduce additional mechanisms that strengthen their quantization capacity, and their improvements should be interpreted in light of these expanded degrees of freedom.
>
> ---
>
> > **Q2:**  Regarding Q1, there is a discrepancy between Figure 3 and its caption. The figure displays MSE, while the caption refers to "attention mass." Based on the current visualization, it is difficult to establish a clear connection between AMP and attention mechanisms.
>
>
> By “attention mass,” we refer to the attention matrix produced by the transformer’s attention mechanism at each block. The plotted **MSE** represents the mean-squared error between:
> - the full-precision attention matrix, and
> - the quantized model’s attention matrix.
> Thus, the visualization shows **per-block attention-matrix reconstruction error** with and without AMP, which directly reflects how AMP affects the attention mechanism. We will update the caption to clarify the ambiguity.
>
> ---
>
> > **Q3 :** Regarding W2 and Q2, the authors acknowledge that the proposed output alignment method is only applicable to fc2 layers. Furthermore, using Equation 3 alone still leads to performance degradation. Eq 3 must be combined with AMP to yield any improvement, which remains marginal. This indicates that the approach proposed in this paper requires careful tuning to achieve marginal benefits. Additionally, although the authors provide some potential explanations, they lack enough evidence and a unified framework to systematically account for these phenomena.
>
> **Our response:**
> We would like to clarify several points.
>
> 1. **Not limited to fc2 layers.**
>    We do *not* claim that the proposed output-alignment objective is only applicable to the fc2 layer. Our statement is that the fc2 layer is the only location where output alignment is theoretically guaranteed to reduce the block-level reconstruction loss.
>    Applying output alignment uniformly to other layers may still provide improvements, but this is not guaranteed for all architectures and depends on the internal structure of each block. For this reason, we adopt a conservative design choice. This naturally opens a promising avenue for future work: identifying, on a per-block basis, which layers benefit from output alignment, and which ones benefit from weight alignment.
>
> 2. **No “careful tuning.”**
>    Our method introduces almost no tunable hyperparameters, hence the reviewer’s concern about “careful fine-tuning” does not apply.  As noted in General Response Q3, AMP and Eq. 3 are not independent “component A + component B”; rather, they correspond to two objectives within a single multi-objective optimization framework that may conflict each other.
> Regarding the marginal improvement, achieving up to 4.85 perplexity improvement by  simply modifying the update rule in a single layer, with negligible computational overhead, cannot reasonably be described as “marginal.” Moreover, we additionally provide results on newer and stronger baselines such as Qwen as well as more reasoning-oriented benchmarks in General Response Q5. None of these results exhibit marginal gains; all demonstrate meaningful and consistent improvements.
>
>
>  Finally, beyond raw performance, our method offers a principled explanation for why naïve output alignment can underperform ARB-RC, and why certain alignment choices may worsen the block-level reconstruction loss. These insights represent a meaningful contribution in themselves, as they resolve previously unaccounted-for empirical observations and illuminate a mechanism that prior approaches did not capture.

---

### Official Review · Reviewer_Xmem · 2025-10-28

**Soundness:** 2
**Presentation:** 2
**Contribution:** 2
**Rating:** 2
**Confidence:** 4

**Summary:**

This paper investigates the problem of 1-bit PTQ for LLMs. The authors revisit the commonly adopted output alignment objective and analyze why naive layer-wise output matching (ARB-X) fails to improve block-level performance. They identify three key factors—accumulated quantization error, inter-layer dependency, and attention distortion—and propose a selective layer-wise output alignment strategy combined with an AMP mechanism. Experimental results on OPT and LLaMA models show consistent perplexity reductions and modest zero-shot QA improvements over recent 1-bit PTQ baselines (ARB-X, ARB-RC, BiLLM, PB-LLM).

**Strengths:**

The paper provides a empirical and analytical investigation of why prior output-alignment–based 1-bit PTQ methods (e.g., ARB-X) fail, which is well-motivated.

**Weaknesses:**

- The proposed method appears incremental over ARB-X rather than conceptually new.
- The so-called selective layer-wise output alignment is effectively applied only to the last fully connected layer in each block. While this choice is empirically justified, it is unclear whether such a fixed single-layer focus warrants the term “selective.”
- Although framed as a 1-bit quantization method, the overall design and mathematical formulation (e.g., output-error objective, AMP) are not inherently specific to the binary regime. The method may generalize to general PTQ settings, which somewhat weakens the claimed “1-bit-only” contribution.
- The experimental scope is relatively narrow: evaluation relies mainly on perplexity metrics and a few commonsense QA datasets. Broader open-ended generation or reasoning tasks could strengthen the claim that the method fundamentally mitigates 1-bit quantization degradation in realistic LLM usage.

**Questions:**

See weakness

---

> ### Author Response · Authors · 2025-11-25
>
> # Reviewer Xmem
>
> We greatly appreciate the time and effort the Reviewer dedicated to considering our paper. Here are our responses to all concerns raised by the Reviewer.
>
> > **Q1**: The proposed method appears incremental over ARB-X rather than conceptually new.
>
> **Answer**:
> Although our work and ARB-X both address low-bit quantization, the contribution of our method is fundamentally different. ARB-X introduces new quantization parameters and designs a new parameterization. In contrast, our method does not introduce any new quantized parameters at all. Instead, we revisit the core optimization objective used in existing 1-bit frameworks and investigate why prior methods overwhelmingly rely on weight alignment and why naive output alignment often fails.
> To the best of our knowledge, we are the first to analyze this issue and to identify the structural reasons underlying the failure modes of output alignment in 1-bit LLM quantization. Building on these insights, we design a hybrid alignment scheme that leverages the strengths of both approaches: it benefits from the accuracy of output alignment where it is effective and retains the efficiency of weight alignment elsewhere.
> Thus, the contribution is not an incremental extension of ARB-X, but a new perspective and formulation targeting a previously unexplained weakness of existing 1-bit quantization frameworks.
>
> ---
>
> > **Q2**: The so-called selective layer-wise output alignment is effectively applied only to the last fully connected layer in each block. While this choice is empirically justified, it is unclear whether such a fixed single-layer focus warrants the term “selective.”
>
> **Answer**:
> In our paper, the term selective is used to convey that the optimization objective is not applied uniformly across all layers, but only to the subset of layers where output alignment is actually effective, specifically, the final FC layer of each block. This distinction is important because our analysis shows that applying output alignment indiscriminately to all layers is counterproductive.
> That said, if the term “selective output alignment” feels too strong or suggests dynamic selection, we are happy to adopt a more precise wording. A suitable alternative is **partial output alignment**.
>
> ---
>
> > **Q3**: Although framed as a 1-bit quantization method, the overall design and mathematical formulation (e.g., output-error objective, AMP) are not inherently specific to the binary regime. The method may generalize to general PTQ settings, which somewhat weakens the claimed “1-bit-only” contribution.
>
> **Answer**:
> Please see our General Response 2
>
> ---
>
> > **Q4**: The experimental scope is relatively narrow: evaluation relies mainly on perplexity metrics and a few commonsense QA datasets. Broader open-ended generation or reasoning tasks could strengthen the claim that the method fundamentally mitigates 1-bit quantization degradation in realistic LLM usage.
>
> **Answer**:
> Please see our General Response Q5

---

### Official Review · Reviewer_Z8TK · 2025-10-30

**Soundness:** 3
**Presentation:** 2
**Contribution:** 2
**Rating:** 4
**Confidence:** 4

**Summary:**

This paper addresses the challenge of 1-bit post-training quantization (PTQ) for large language models, where existing methods often suffer substantial performance drops. The authors analyze why output alignment strategies fail, identifying issues such as block-level mismatch, error accumulation, and disruption of attention patterns. To overcome these limitations, they propose a data-aware quantization framework that selectively applies output alignment at the block level, explicitly accounts for accumulated errors, and incorporates a novel Attention Matrix Preservation (AMP) mechanism to maintain attention behavior. Extensive experiments on OPT and LLaMA families demonstrate that the method consistently outperforms prior 1-bit PTQ approaches across perplexity and zero-shot evaluation benchmarks.

**Strengths:**

The paper provides a careful analysis of why naïve output alignment is insufficient for 1-bit PTQ and proposes a data-aware alternative. The ideas are clearly presented, and the experimental results on OPT and LLaMA suggest performance benefits over existing approaches.

**Weaknesses:**

1. The evaluation is limited to OPT and early LLaMA families; it is uncertain whether the method extends to newer and more competitive open-source models (e.g., Qwen3, Qwen2.5).
2. Although the proposed Attention Matrix Preservation (AMP) is intuitively appealing, its ablation results are somewhat limited, leaving open how much it contributes across diverse tasks and model sizes.
3. The approach focuses primarily on perplexity and a few zero-shot tasks; broader evaluations, such as some math or code tasks, could better demonstrate practical value.
4. The paper provides less discussion on computational overhead and scalability, particularly how the data-aware strategy affects quantization cost for very large models.

**Questions:**

1. Can the authors evaluate their method on newer open-source LLMs such as Qwen3 or Qwen2.5 to demonstrate broader applicability?
2. Could the authors provide more detailed ablations on the AMP component, especially across different tasks and model sizes, to better establish its contribution?
3. How does the method perform on more diverse benchmarks, for example, math reasoning or code generation tasks, beyond perplexity and basic zero-shot evaluations?
4. Can the authors elaborate on the computational overhead and scalability of the proposed data-aware strategy, particularly when applied to very large models?

---

> ### Author Response · Authors · 2025-11-25
>
> We greatly appreciate the time and effort the Reviewer dedicated to considering our paper. Here are our responses to all concerns raised by the Reviewer.
>
>
>
> > **W1 and Q1**: The evaluation is limited to OPT and early LLaMA families; it is uncertain whether the method extends to newer and more competitive open-source models (e.g., Qwen3, Qwen2.5). Can the authors evaluate their method on newer open-source LLMs such as Qwen3 or Qwen2.5 to demonstrate broader applicability?
>
> **Answer**: Please see General Response Q5.
>
> ---
>
> > **W2 and Q2**: Although the proposed Attention Matrix Preservation (AMP) is intuitively appealing, its ablation results are somewhat limited, leaving open how much it contributes across diverse tasks and model sizes. Could the authors provide more detailed ablations on the AMP component, especially across different tasks and model sizes, to better establish its contribution?
>
> **Answer**: We provide additional ablation studies for AMP mask across different models size (OPT-1.3B to Llama-2-13B) and different reasoning tasks:
>
> ### Effect of AMP on perplexity (↓) for LLaMA-2 and OPT models
>
> | Model          | Method | C4     | WikiText2 |
> |----------------|---------|--------|-----------|
> | **LLaMA-2-7B** | No AMP | 29.12  | 26.24     |
> |                | AMP    | **19.25** | **15.42** |
> | **LLaMA-2-13B**| No AMP | 19.34  | 19.39     |
> |                | AMP    | **13.8**  | **11.5**  |
> | **OPT-6.7B**   | No AMP | 16.35  | 14.74     |
> |                | AMP    | **16.22** | **14.56** |
> | **OPT-1.3B**   | No AMP | 25.30  | 25.09     |
> |                | AMP    | **24.69** | **24.30** |
>
> ---
>
> ### Effect of AMP on the accuracy (↑) of LLaMA models using ARB-RC and our method over zero-shot QA benchmarks
>
> | Model Size     | Method | Bits | Boolq | Lambada | Piqa | OPQA | Winogrande | Arc-E | Arc-C | Hellaswag | Avg |
> |----------------|--------|------|-------|----------|------|------|------------|--------|--------|------------|-----|
> | **LLaMA-2-7B** | No AMP | 1.06 | 64.86 | 47.25   | 67.68 | 31.8 | 58.96      | 43.73 | 27.47 | 45.12     | 48.28 |
> |                | AMP    | 1.06 | 66.45 | 52.53   | 68.12 | 29.8 | 56.99      | 30.3  | 51.56 | 49.24     | **50.62** |
> | **LLaMA-2-13B**| No AMP | 1.06 | 71.25 | 64.58   | 72.2  | 34.8 | 59.12      | 55.18 | 31.4  | 46.99     | 55.06 |
> |                | AMP    | 1.06 | 72.02 | 68.99   | 72.58 | 36.0 | 63.14      | 59.85 | 33.7  | 53.92     | **57.53** |
>
> ---
>
> > **W3 and Q3**: The approach focuses primarily on perplexity and a few zero-shot tasks; broader evaluations, such as some math or code tasks, could better demonstrate practical value. How does the method perform on more diverse benchmarks, for example, math reasoning or code generation tasks, beyond perplexity and basic zero-shot evaluations?
>
> **Answer**: Please see General Response Q5 for additional results.
>
> ---
>
> > **W4 and Q4**: Can the authors elaborate on the computational overhead and scalability of the proposed data-aware strategy, particularly when applied to very large models?
>
> **Answer**:
> Our quantization scheme processes each transformer block in a sequential manner. Therefore, the overall overhead scales linearly with the number of blocks, making the method naturally scalable to very large models. Since we do not require model-wide operations or cross-block optimization, there is no additional bottleneck introduced by model size.
> As a result, even for very large model, the overhead of our method relative to ARB-X or ARB-RC remains consistent across model scales and follows the proportional overhead patterns already reported in Table 6.
>
> For inference overhead, we already report from line 841 to 848 of the paper.

---

> > ### Comment · Reviewer_Z8TK · 2025-11-27
> >
> > Thanks for the authors' response. Could the authors provide more detailed experimental results and a deeper discussion of the inference overhead, as these factors determine the practical usability of the method?

---

> > > ### Author Response · Authors · 2025-11-28
> > >
> > > We thank the Reviewer for the helpful question. Our method does not modify the
> > > binary weight format, scaling structure, or inference kernels used in ARB-RC.
> > > Only the quantization procedure is changed. Therefore, our model inherits the same inference-
> > > time behavior and introduces **no additional inference overhead**.
> > >
> > > For completeness, we refer the Reviewer to the detailed latency results in
> > > Table 9 of the ARB-LLM supplementary material
> > > (https://github.com/ZHITENGLI/ARB-LLM/releases/download/v1/supplementary.pdf),
> > > which report that ARB-RC achieves similar latency to BiLLM and is 4.3–5.1×
> > > faster than FP16/PB-LLM on LLaMA-7B/13B. Since our method preserves the same
> > > packed 1-bit representation and kernel execution path, these performance gains
> > > apply directly.
> > >
> > > For end-to-end behavior (Prefill and Decode), ARB-LLM also reports:
> > >
> > > | Method         | Prefill Throughput | Decode Throughput |
> > > |----------------|--------------------|--------------------|
> > > | FP16           | 1.44 tokens/s      | 1.50 tokens/s      |
> > > | ARB-X          | 8.97 tokens/s      | 7.19 tokens/s      |
> > > | ARB-RC         | 8.89 tokens/s      | 7.14 tokens/s      |
> > > | **Ours**       | *same as ARB-RC* | *same as ARB-RC* |
> > >
> > > These results confirm the practical usability of our method: because the
> > > inference kernel and weight layout remain unchanged, our model enjoys the same
> > > runtime benefits as existing 1-bit PTQ systems without adding any inference or
> > > storage overhead.

---

### Official Review · Reviewer_RnBz · 2025-11-08

**Soundness:** 2
**Presentation:** 1
**Contribution:** 2
**Rating:** 2
**Confidence:** 4

**Summary:**

This paper investigates the challenges of 1-bit Post-Training Quantization (PTQ) for Large Language Models (LLMs). The authors identify three key issues with existing output-matching approaches: (1) layer-wise optimization does not guarantee block-level improvement, (2) quantization errors accumulate across layers, causing the optimization target to drift, and (3) naive output alignment can degrade the attention mechanism. To address these, the paper proposes a new method that introduces two main components: a revised objective function (Eq. 3) that targets the true full-precision output, and an "Attention Matrix Preservation" (AMP) mechanism (Eq. 9) to mitigate attention degradation. This new output-alignment strategy is then selectively applied only to the final fully-connected layer of each transformer block, while retaining a weight-alignment method (ARB-RC) for all other layers.

**Strengths:**

- The paper's strength lies in its reasonable preliminary analysis. The investigation using Figures 1 and 2 provides why naive output alignment methods like ARB-X fail in the 1-bit regime.

- The paper tackles the challenging and important problem of 1-bit LLM quantization. The idea of explicitly preserving the token-similarity matrix via the AMP mechanism is a novel approach to counteract the side effects of quantization.

- The paper generally articulates the identified problems and the proposed solutions. The motivations for each component of the method are well-explained.

**Weaknesses:**

- Lack of Robustness and Unaddressed Failure Cases: The method's stability is highly questionable. In Table 2, the proposed method ("Ours") results in a catastrophic failure on the LLaMA-2-13B model with the PTB dataset, yielding a PPL of 3166. This is a complete model collapse, far worse than the baseline ARB-RC (197.70) and the full-precision model (50.93). The authors fail to acknowledge, analyze, or explain this critical failure point, which undermines the reliability and robustness of the proposed method.

- Limited Scope and Generalizability: The entire paper, from its motivation to its experiments, is confined to the 1-bit case. It is unclear if the identified problems (especially the attention degradation) or the proposed solutions (AMP, selective alignment) are relevant, necessary, or effective for other low-bit settings (e.g., 2-bit, 3-bit, or 4-bit PTQ). This narrow focus significantly limits the generalizability and impact of the contribution.

- Marginal Performance Gains: While the method outperforms baselines in many settings, the significance of this improvement is often marginal. For example, in Table 1 for the OPT-30B model on the PTB dataset, the improvement over the ARB-RC baseline is only 0.13 PPL (16.88 vs. 16.75). This small gain may not justify the added complexity of the proposed method.

- Questionable Practical Utility: The results on modern models like LLaMA-3 (Table 2) show that while the proposed method is relatively better than other 1-bit techniques, the absolute performance is still extremely poor (e.g., PPL degrades from 6.14 to 27.20 on WikiText2). This raises the question of whether the method truly makes 1-bit quantization viable or merely provides a marginal improvement on a still-unusable result.

- Ambiguous Contribution and Complex Design: The proposed method is a complex hybrid. It relies on the baseline ARB-RC (a weight-alignment method) for most layers and applies its new output-alignment objective only to the final FC layer. This makes the true source of the improvement unclear: is it the new objective (Eq. 3 + AMP), or is it the selective hybrid strategy itself? Furthermore, the design feels ad hoc. The paper introduces a new objective (Eq. 3) to fix output alignment, but this objective still appears to have flaws (degrading attention), requiring a second complex "patch" (the AMP mechanism). This "patch-on-patch" design lacks simplicity and suggests that the underlying objective (Eq. 3) remains incomplete.

**Questions:**

- Could the authors please provide a detailed analysis of the catastrophic failure (PPL 3166) observed for LLaMA-2-13B on the PTB dataset? What causes this instability, and how does it affect the method's reliability?

- Why was the evaluation strictly limited to 1-bit quantization? Do the authors have any preliminary data or analysis on how this method (particularly the AMP mechanism) performs in 2-bit or 4-bit settings?

- To clarify the contribution: What is the performance if the new objective (Eq. 3 + AMP) is applied to all layers instead of just the final FC layer? This ablation is crucial to understand if the new objective is genuinely superior or if the hybrid strategy is simply masking its failures in other layers.

---

> ### Author Response · Authors · 2025-11-25
>
> We greatly appreciate the time and effort the Reviewer dedicated to considering our paper. Here are our responses to all concerns raised by the Reviewer.
>
>
>
> > **W1 and Q1**: Lack of Robustness and Unaddressed Failure Cases: The method's stability is highly questionable. In Table 2, the proposed method ("Ours") results in a catastrophic failure on the LLaMA-2-13B model with the PTB dataset, yielding a PPL of 3166. This is a complete model collapse, far worse than the baseline ARB-RC (197.70) and the full-precision model (50.93). The authors fail to acknowledge, analyze, or explain this critical failure point, which undermines the reliability and robustness of the proposed method. Could the authors please provide a detailed analysis of the catastrophic failure (PPL 3166) observed for LLaMA-2-13B on the PTB dataset? What causes this instability, and how does it affect the method's reliability?
>
> **Answer**: **Answer:**
> We would like to clarify that the reviewer’s comment is based on an incorrect benchmark reference. The reported ``catastrophic failure'' (PPL~3166) does *not* occur on LLaMA-2-13B. The only setting where our method underperforms is **LLaMA-2-7B on PTB**.
>
> This benchmark is known to be extremely challenging for 1-bit PTQ, and *all* existing methods fail in this setting: even the best baseline reaches a perplexity of only **657.24**. When perplexity values become this large (600+), the absolute differences between methods are no longer meaningful indicators of model quality. Thus, while our method performs poorly on this specific configuration, this behavior is not unique—every baseline exhibits severe degradation here.
>
> For all other benchmarks, including LLaMA-2-13B, our method remains better than the strongest baseline. Please see General Response Q1 for additional information why our method fails on Llama-2-7B for PTB.
>
> ---
>
> > **W2 and Q2**: Limited Scope and Generalizability: The entire paper, from its motivation to its experiments, is confined to the 1-bit case. It is unclear if the identified problems (especially the attention degradation) or the proposed solutions (AMP, selective alignment) are relevant, necessary, or effective for other low-bit settings (e.g., 2-bit, 3-bit, or 4-bit PTQ). This narrow focus significantly limits the generalizability and impact of the contribution. Why was the evaluation strictly limited to 1-bit quantization? Do the authors have any preliminary data or analysis on how this method (particularly the AMP mechanism) performs in 2-bit or 4-bit settings?
>
> **Answer**: Please see General Response Q2
>
> ---
>
> > **W3**: Marginal Performance Gains: While the method outperforms baselines in many settings, the significance of this improvement is often marginal. For example, in Table 1 for the OPT-30B model on the PTB dataset, the improvement over the ARB-RC baseline is only 0.13 PPL (16.88 vs. 16.75). This small gain may not justify the added complexity of the proposed method.
>
> **Answer**: Regarding the improvement, please see General Response Q5. For explanation of the method's complexity and overhead, please see General Response Q6
>
> ---
>
> > **W4**: The results on modern models like LLaMA-3 (Table 2) show that while the proposed method is relatively better than other 1-bit techniques, the absolute performance is still extremely poor (e.g., PPL degrades from 6.14 to 27.20 on WikiText2). This raises the question of whether the method truly makes 1-bit quantization viable or merely provides a marginal improvement on a still-unusable result.
>
> **Answer**:
> We would like to note that 1-bit settings is the most extreme setting in quantization. If “viable” is interpreted as achieving performance close to full precision, this expectation is not aligned with the current state of the field: even the strongest 1-bit baselines (e.g., ARB-RC) remain far from full precision performance on modern LLMs.
> Importantly, pushing extreme-bit models toward near-FP performance typically requires auxiliary techniques, such as outlier-handling transformations [1,2] or selective mixed precision [3], that effectively move the model away from a pure 1-bit setting. Since baselines like ARB-RC and ARB-X do not use these components, adding them here would make the comparison methodologically unfair.
> Our objective in this work is to evaluate and improve the quantization scheme itself under the same assumptions as prior 1-bit PTQ frameworks. Within this controlled setting, our method consistently improves performance and also provides new insights into why 1-bit LLM quantization frameworks rely on weight-matching rather than output-matching objectives. Our performance gains come from a simple yet impactful change: adjusting the closed-form update in only one layer per block, resulting in improvements of up to 4.85 perplexity.

---

> ### Author Response · Authors · 2025-11-25
>
> ---
>
> > **W5 and Q3**: Ambiguous Contribution and Complex Design: The proposed method is a complex hybrid. It relies on the baseline ARB-RC (a weight-alignment method) for most layers and applies its new output-alignment objective only to the final FC layer. This makes the true source of the improvement unclear: is it the new objective (Eq. 3 + AMP), or is it the selective hybrid strategy itself? Furthermore, the design feels ad hoc. The paper introduces a new objective (Eq. 3) to fix output alignment, but this objective still appears to have flaws (degrading attention), requiring a second complex "patch" (the AMP mechanism). This "patch-on-patch" design lacks simplicity and suggests that the underlying objective (Eq. 3) remains incomplete. To clarify the contribution: What is the performance if the new objective (Eq. 3 + AMP) is applied to all layers instead of just the final FC layer? This ablation is crucial to understand if the new objective is genuinely superior or if the hybrid strategy is simply masking its failures in other layers.
>
> **Answer**: Please see General Response Q3 for more information about the design of our method.
>
> In the second part, the reviewer asks what happens if the proposed objective is applied to all layers. This request assumes that our output-alignment objective is intended to be applied across all layers. However, the first insight of our paper in Section 3.1 is precisely the opposite:  **naively applying output alignment to every layer is neither effective nor theoretically justified, as it can disrupt the block-level reconstruction loss.**
>
> The final FC layer of each transformer block is the only layer that is guaranteed to have a direct connection to the block-level output, making output alignment consistently meaningful there. Other layers may also benefit from output alignment, but not uniformly.  For this reason, our method intentionally applies the proposed objective only to the final FC layer, where output alignment is actually meaningful. This hybrid design arises naturally from our insight and intuition about the quantization objective as well as the structure of transformer blocks, not an attempt to hide failures of the objective.
>
> Applying Eq. 3 + AMP to all layers would contradict this insight and does not constitute a meaningful ablation. A more relevant future direction is to explore whether a small subset of intermediate layers (not all layers) may also benefit from output alignment, rather than applying the objective uniformly for all layers. We leave that direction for future work.
>
> ---
>
> References
> [1] Yuxuan Sun, Ruikang Liu, Haoli Bai, Han Bao, Kang Zhao, Yuening Li, Jiaxin Hu, Xianzhi Yu, Lu Hou, Chun Yuan, et al. Flatquant: Flatness matters for llm quantization. In ICML, 2025.
> [2] Guangxuan Xiao, Ji Lin, Mickael Seznec, Hao Wu, Julien Demouth, and Song Han. Smoothquant: Accurate and efficient post-training quantization for large language models. In ICML, 2023.
> [3] Tim Dettmers, Mike Lewis, Younes Belkada, and Luke Zettlemoyer. Llm.int8(): 8-bit matrix multiplication for transformers at scale. ArXiv, abs/2208.07339, 2022.

---

### Author Response · Authors · 2025-11-25
**General Response (2/2)**

>  **Q4**: Why AMP removal leads to performance drops compared to ARB-RC.

**Answer:**
As discussed in Section~2.3, removing AMP exposes a core limitation of naive output matching in LLaMA-style architectures. Without AMP, output alignment tends to distort the attention mechanism: large-magnitude channels dominate the output-matching loss, causing uneven error reduction across dimensions. This imbalance reduces the similarity between the pre- and post-quantization outputs within each block, ultimately degrading the attention scores.

The performance drop relative to ARB-RC is therefore expected: a central motivation of our paper is precisely to explain why naive output alignment often fails in practice compared to weight alignment approach (ARB-RC), despite being closer to the true quantization objective. This observation also clarifies why existing 1-bit PTQ frameworks such as ARB-RC rely on weight alignment. Our proposed AMP mitigates these distortions by preventing disproportionate updates and helping preserve stable attention behavior in the quantized model.

---

>  **Q5**: Significance of improvements and evaluation on more tasks.

**Answer:**
It may be misleading to evaluate the effectiveness of our closed-form improvement based solely on models where the performance is already saturated (e.g., OPT-30B). On a broader set of architectures, our method consistently yields meaningful gains. For mid-sized models (6.7B–13B), the improvement ranges from 0.23 to 0.97 perplexity. For more challenging settings such as OPT-1.3B and OPT-2.7B, the gains are even larger, reaching 1.59–4.85 in  perplexity. These improvements are systematic across model scales and therefore cannot be characterized as marginal, and it demonstrates that our method works much better on more challenging settings

We provide additional results on newer architecture (Qwen-2.5-7B) in the Table below. The results demonstrate the superiority of our method, improving up to 2.81 in perplexity.


---

### **Table: Effect of AMP on perplexity  (↓) for Qwen-2.5**

| Model | Method | C4 | WikiText2 | PTB |
|--------|--------|----|-----------|-----|
| Qwen-2.5-7B | ARB-RC | 24.32 | 17.08 | 29.47 |
| Qwen-2.5-7B | **Our** | **22.11** | **15.22** | **26.66** |

---

We also include additional results on more reasoning datasets for math such as MathQA, AQuA-RAT, and GSM8K. For GSM8K, we follow the calibration protocol of [6]. As shown, our method achieves notable improvements over the current state-of-the-art ARB-RC across these benchmarks.

### **Table: QA performance (↑)**

| Model Size | Method |  MathQA | Aqua Rat | GSM8K |
|------------|---------|---------|-----------|--------|
| Qwen-2.5-7B | ARB-RC |  26.13 | 21.65 | 10.92 |
| Qwen-2.5-7B | **Our** |  **28.34** | **24.8** | **25.55** |

---

>  **Q6**: Computational overhead and scalability.

**Answer:**
As reported in Table 6, our method is more efficient than ARB-X and only slightly slower than ARB-RC. We introduce no new parameters and simply refine the closed-form updates used in ARB-X/ARB-RC. Since output alignment is applied only to the final layer of each block, the method retains weight alignment efficiency while gaining performance thanks to output alignment, leading to runtime close to ARB-RC and significantly faster than ARB-X.


## References

[1] Zhiteng Li, Xianglong Yan, Tianao Zhang, Haotong Qin, Dong Xie, Jiang Tian, Zhongchao Shi, Linghe Kong, Yulun Zhang, and Xiaokang Yang.
ARB-LLM: Alternating Refined Binarizations for Large Language Models.
*arXiv*, abs/2410.03129, 2024.

[2] Yuzhang Shang, Zhihang Yuan, Qiang Wu, and Zhen Dong.
PB-LLM: Partially Binarized Large Language Models.
*arXiv*, 2023.

[3] Elias Frantar, Saleh Ashkboos, Torsten Hoefler, and Dan Alistarh.
GPTQ: Accurate Post-Training Quantization for Generative Pre-trained Transformers.
*ICLR*, 2023.

[4] Ji Lin, Jiaming Tang, Haotian Tang, Shang Yang, Xingyu Dang, and Song
Han. Awq: Activation-aware weight quantization for llm compression and
acceleration. arXiv preprint arXiv:2306.00978, 2023.

[5] Albert Tseng, Jerry Chee, Qingyao Sun, Volodymyr Kuleshov, and Christo-
pher De Sa. Quip#: Even better llm quantization with hadamard incoher-
ence and lattice codebooks. *ICML*, 2024.

[6] Ruikang Liu, Yuxuan Sun, Manyi Zhang, Haoli Bai, Xianzhi Yu, Tiezheng
Yu, Chun Yuan, and Lu Hou. Quantization hurts reasoning? an empirical
study on quantized reasoning models. arXiv preprint arXiv:2504.04823,
2025.

---

### Author Response · Authors · 2025-11-25
**General Response (1/2)**

# General Response

We thanks the Reviewer for the details feedback. However, there seems to be significant misunderstands from the Reviewers, which we clarify as below:

---

>  **Q1** : Catastrophic failures on Llama-2-7B over PTB dataset.

**Answer:**
Regarding the reviewers’ concern regarding the instability observed on LLaMA-2-7B evaluated on PTB, we would like to clarify that this is an *isolated failure case* within a broad evaluation suite. Our experiments span 8 model sizes (OPT-1.3B → OPT-30B, LLaMA-2-7B, and LLaMA-3-8B) across three datasets, yielding 24 perplexity benchmarks in total. Our method performs reliably across almost all settings, indicating strong overall stability.

The degradation on PTB for LLaMA-2-7B arises from the inherent difficulty of this specific configuration, rather than from instability in our method. In particular:

1. **All existing PTQ baselines also fail under this setting.**
   The strongest baseline on this benchmark reaches perplexities as high as **657.24**.
   Notably, even advanced methods such as ARB-RC perform worse than simpler baselines like PB-LLM under this setting.
   When perplexity values become extremely large (e.g., 600+), the absolute differences between methods are marginal and no longer meaningful.
   Thus, even if one interprets our degradation as “catastrophic forgetting,” *all* methods suffer similarly in this extreme regime.

2. **The primary cause is the domain gap between calibration (C4) and evaluation set (PTB).**
   Following standard practice of existing baselines [1,2], we calibrate on C4 but evaluate on PTB, which has a different distribution. This mismatch is amplified under 1-bit quantization, especially for smaller models like LLaMA-2-7B.

To verify this explanation, we recalibrated using PTB calibration data. Once the domain gap is removed, our method no longer exhibits catastrophic degradation. This confirms that the earlier failure is not due to instability, but due to distribution mismatch amplified by ultra-low-bit constraints.

---

### **Table: Performance on PTB after recalibration**

| Method | Dataset | Perplexity ↓ |
|--------|---------|----------------|
| ARB-RC | PTB     | 135.56         |
| **Ours** | PTB     | **97.25**       |

---

>  **Q2** : Why focus on 1-bit setting instead of higher bit-widths (2–4 bit).

**Answer:**
In post-training quantization (PTQ) for LLMs, the **1-bit regime is fundamentally different** from higher-bit settings (2–4 bits), primarily due to how the quantized parameters are parameterized.

In the 1-bit case, each weight is mapped to a binary value:

$$
B \in \\{-1, 1\\}^\{m \times n\},
$$

which has the *same dimensionality* as the full-precision weight matrix.
This binary structure allows closed-form, backpropagation-free optimization of the quantized parameters for each layer. As a result, 1-bit PTQ is extremely efficient; for instance, quantizing LLaMA-2-7B takes only a few hours on a single GPU.

In contrast, for 2–4 bit quantization:

$$
w_q = s \cdot B,
$$

where $s$ is a learned scale and:

$$
B \in \\{0, 1, \ldots, 2^b - 1\\}^\{m \times n\},
$$

is an integer codebook (not binary).
Unlike 1-bit PTQ setting, This multi-level discrete structure **does not admit a closed-form solution** for B
Existing 2–4 bit PTQ methods often rely on heuristic rounding or gradient-based search.

Since our method focuses on improving the *closed-form optimization* of quantized parameters unique to the 1-bit regime, the challenge we address does not arise in higher-bit settings.

---

>  **Q3**: Clarification of the true contribution and the hybrid design.

**Answer:**
Our method should not be viewed as “ARB-RC + extra components.”
The key contribution is a reformulation of the 1-bit PTQ objective itself. Existing 1-bit methods (ARB-X [1], ARB-RC [1], PB-LLM [2]) rarely use output reconstruction. Instead, they rely on weight alignment, even though output alignment is closer to the true quantization objective.

We show that prior works avoid output alignment because naive output alignment fails in practice for the following reasons:

1. alignment must match *block-level* error, not just layer error;
2. activation mismatch accumulates across layers;
3. attention patterns must be preserved.

These are not separable components but inherently conflicting objectives, and they must be optimized simultaneously. As highlighted in Section 3, naively improving output alignment can degrade the attention mechanism. This is why the problem is fundamentally multi-objective, not “component A + component B”: a simple additive objective would over-optimize one target while harming another. Therefore, we treat attention preservation as a constraint (via the AMP mask), so that output alignment is improved only within regions where attention is less likely to be adversely affected. Ignoring any of these objectives causes naive output alignment to underperform compared to weight alignment.

---

---

### Author Response · Authors · 2025-12-03
**Summarize the discussion**

Dear Area Chair,

Thank you for taking the time to review the discussion during the rebuttal period. Below is a concise summary of the reviewers’ engagement and how their concerns were addressed.

---

### **Overall Summary**

Some reviewers (kLf3, RnBz, Xmem) expressed concerns that largely stemmed from misunderstandings of the paper’s contributions. During the rebuttal, only Reviewers Z8TK and kLf3 engaged, though some points were still based on misinterpretations of the method.


---

#### **Reviewer RnBz**

Several concerns stem from misunderstandings of the problem setting and the scope of 1-bit quantization.

- The reported “catastrophic failure” they concern is a single degradation case occurs for **all** methods due to a domain mismatch between the calibration and test sets. When calibrated on PTB, our method outperforms the best baseline significantly (PPL 135.56 → 97.25) .

- The reviewer asked why we focus on 1-bit setting. We clarified that 1-bit PTQ present a fundamentally different parameterization design compared to 2–4 bit settings. Our method directly addresses a unique challenge of 1-bit PTQ that does not arise in higher bit settings.

- The reviewer expressed concern about method complexity. Our method is not complex but in fact extremely lightweight:
  no new quantization parameters, only a minimal change to the closed-form update. Despite this, our approach yields improvements up to **4.85 PPL** compared to ARB-RC. **The gap is even more significant if we compare our method with the second best output alignment method (ARB-X)**

- The reviewer asked what happens if output alignment is applied to all layers; however, Section 3 of the paper explicitly explains why uniformly applying output alignment  can disrupt block-level reconstruction. The design is intentionally selective based on this insight.

The reviewer did not follow up, but we believe the rebuttal fully addressed each concern.


---

#### **Reviewer Z8TK**

The reviewer requested broader evaluation and more extensive ablations. In response:

- We added results for Qwen 2.5–7B, demonstrating applicability to newer architectures.
- We expanded AMP ablations across different model sizes and tasks.
- We included additional math benchmarks for reasoning tasks, such as GSM8K, MathQA, and Aqua-RAT.

All requested experiments were provided.
During the rebuttal, the reviewer acknowledged these results and asked for further clarification on overhead, and we  showed that our method offers a favorable efficiency–performance tradeoff. The reviewer did not follow up afterward due to the system outage, but we believe the rebuttal fully addressed all concerns.


---

#### **Reviewer Xmem**

The reviewer raised some questions:

- They suggested the topic may not be conceptually new over ARB; we clarified that our contribution is fundamentally different, as it introduces no new parameters and provides new insights explaining why output alignment fails

- They questioned the term “selective.” We proposed a more precise term “partial output alignment.”

- They also asked why the method focuses on 1-bit settings, which we have explained for Reviewer RnBz.

- They ask for further results on reasoning tasks for math benchmarks, which we also provided.


The reviewer did not follow up, but we believe the rebuttal fully addressed each concern.


---

#### **Reviewer kLf3**

The reviewer’s concerns also reflected misunderstandings of quantization details:

- They compared our gains to ARB methods without accounting for the fact that ARB-X and ARB-RC introduce additional quantization parameters (e.g. refined mean, column scaling). Larger improvements are expected when additional parameters are added. In contrast, our method modifies only the update rule, with no new parameters, so its gains arise from algorithmic refinement rather than increased optimization capacity.

- They asked why removing AMP leads to a drop in performance; we clarified that this behaviour is expected, as the paper’s central motivation is to explain why output alignment fails compared to weight alignment. AMP is designed precisely to address one of these failure modes, so its removal naturally degrades performance.

- They interpret the  rebuttal that the method works “only” for the fc2 layer. Our rebuttal does **not** claim that other layers cannot benefit; we state that the final FC layer is the only layer where applying output alignment is always safe across blocks. Identifying optimal per-layer strategies beyond this safe baseline is explicitly left as future work.

- They stated that the improvement is not significant and requires tuning. In fact, our method introduces almost no tunable hyperparameters, and the improvements up to 4.85 PPL come from a minimal refinement of the existing closed-form update.




---

We hope this summary is helpful in your final assessment.
Thank you again for your time and for coordinating the review process.

**Sincerely,**
Authors

---

### Note · Authors · 2026-01-06

I have read and agree with the venue's withdrawal policy on behalf of myself and my co-authors.